# A transcription-based mechanism for oncogenic β-catenin-induced lethality in BRCA1/2-deficient cells

Rebecca A. Dagg [1], Gijs Zonderland [1,5], Emilia Puig Lombardi[1], Giacomo G. Rossetti [2], Florian J. Groelly [1], Sonia Barroso [3], Eliana M. C. Tacconi [1], Benjamin Wright[4], Helen Lockstone[4], Andrés Aguilera [3], Thanos D. Halazonetis[2] & Madalena Tarsounas [1✉]

*BRCA1* or *BRCA2* germline mutations predispose to breast, ovarian and other cancers. High-throughput sequencing of tumour genomes revealed that oncogene amplification and *BRCA1/2* mutations are mutually exclusive in cancer, however the molecular mechanism underlying this incompatibility remains unknown. Here, we report that activation of β-catenin, an oncogene of the WNT signalling pathway, inhibits proliferation of BRCA1/2-deficient cells. RNA-seq analyses revealed β-catenin-induced discrete transcriptome alterations in BRCA2-deficient cells, including suppression of *CDKN1A* gene encoding the CDK inhibitor p21. This accelerates G1/S transition, triggering illegitimate origin firing and DNA damage. In addition, β-catenin activation accelerates replication fork progression in BRCA2-deficient cells, which is critically dependent on p21 downregulation. Importantly, we find that upregulated p21 expression is essential for the survival of BRCA2-deficient cells and tumours. Thus, our work demonstrates that β-catenin toxicity in cancer cells with compromised BRCA1/2 function is driven by transcriptional alterations that cause aberrant replication and inflict DNA damage.

[1] Genome Stability and Tumourigenesis Group, MRC Oxford Institute for Radiation Oncology, Department of Oncology, University of Oxford, Oxford, UK. [2] Department of Molecular Biology, University of Geneva, Geneva, Switzerland. [3] Centro Andaluz de Biología Molecular y Medicina Regenerativa (CABIMER), Universidad de Sevilla-Consejo Superior de Investigaciones Científicas-Universidad Pablo de Olavide, Seville, Spain. [4] Bioinformatics and Statistical Genetics Core, Wellcome Trust Centre for Human Genetics, University of Oxford, Oxford, UK. [5] Present address: Center for Chromosome Stability, Institute for Cellular and Molecular Medicine, Faculty of Health and Medical Sciences, University of Copenhagen, Copenhagen, Denmark. ✉email: madalena.tarsounas@oncology.ox.ac.uk

Deregulated DNA replication is a key contributing factor to the genomic instability characteristic of cancer cells, underlying their ability to proliferate, metastasise and respond to therapy[1]. Aberrant origin firing triggered by oncogene activation and replication fork stalling triggered by barriers that obstruct fork progression (e.g. DNA breaks, DNA secondary structures, DNA–RNA hybrids) represent two prominent examples of pathological replication in cancer.

Germline mutations in *BRCA1* or *BRCA2* tumour suppressor genes predispose individuals to breast, ovarian, prostate and other types of cancer. Functionally, BRCA1 and BRCA2 proteins have physiological roles in maintaining genome integrity, by promoting DNA double-stand break (DSB) repair via homologous recombination and facilitating DNA replication. The latter relies on the ability of BRCA1/2 to re-start, stabilise and protect stalled replication forks against nucleolytic degradation[2]. Consequently, loss of either BRCA1 or BRCA2 activity leads to slow rates of replication fork progression[3,4] and high frequency of stalled forks that are susceptible to degradation by cellular nucleases[5]. When forks fail to re-start and/or are extensively degraded, the replisome disassembles and forks collapse, inflicting DNA damage[1]. Because BRCA1/2 abrogation also inactivates homologous recombination repair, these replication-associated DSBs rely on alternative pathways of repair, most often end joining[6]. By incorrectly re-joining DNA ends, this type of repair leads to mutations and chromosome rearrangements, which fuel genomic instability and tumorigenesis[7].

The critical roles of BRCA1/2 in DNA replication also explain why BRCA-deficient cells and tumours are particularly vulnerable to DNA damage induced by chemotherapy. For example, cross-linking agents such as platinum drugs or alkylating agents induce inter-strand crosslinks[2], whilst G-quadruplex ligands enhance stability of these secondary DNA structures[8], thereby exacerbating the replication defects associated with inactivation of *BRCA1* or *BRCA2* genes. Thus, in cells lacking BRCA1 or BRCA2 replication failure leads to spontaneous DSBs and chromosomal rearrangements. This intrinsic genomic instability represents a hallmark of BRCA-deficient cells and tumours and underlies their sensitivity to chemotherapy that inflicts further DNA damage[5,9].

Similarly to *BRCA1* or *BRCA2* gene mutations, oncogene activation represents an early driver of tumourigenesis, providing cancer cells with selective growth advantage. Consistently, oncogene amplification is one of the most common molecular alterations in cancer, occurring through gene mutation, translocation or amplification[10]. The classical definition of an oncogene is a gene encoding a protein expressed at abnormally high levels, that can transform a normal cell into a cancer cell. In this paper, and consistent with current literature, we refer to the over-expressed protein, as well as the amplified gene encoding it, as oncogenes.

Oncogenes have been shown to deregulate replication and trigger genome instability[11]. Oncogenes interfere with DNA replication via multiple mechanisms including aberrant origin firing, increased transcription and cell cycle progression[12–16], all leading to fork collapse and DSB accumulation[17]. Novel high-throughput technologies, prominently EdU-seq[16], enabled unprecedented mechanistic insight into how oncogenes deregulate DNA replication, in particular the mapping at 10 kb resolution and genome-wide of sites where oncogene-induced origins fire and corresponding forks collapse.

Recently, high-throughput sequencing of breast and ovarian tumours revealed that common oncogene amplifications (i.e. *CCND1* encoding cyclin D1 in breast cancer and *CCNE1* encoding cyclin E in ovarian cancer) rarely occur in tumours with *BRCA1/2* gene mutations[18–20]. The precise mechanism underlying this mutual exclusivity has not been elucidated.

Here, we investigate the effect of oncogenic β-catenin activation in the context of BRCA1/2-deficiency. β-catenin is an oncogene of the WNT pathway frequently overexpressed in cancers, including breast and ovarian[21–23]. Oncogenic β-catenin has been implicated in cancer initiation, growth and metastasis due to its ability to transcriptionally deregulate the expression of multiple genes, with *CCND1* and *MYC* as prominent examples[24,25]. Our results demonstrate that β-catenin activation acts synergistically with *BRCA1* or *BRCA2* abrogation to kill cancer cells. Mechanistically, β-catenin alters the transcription profile of cells lacking BRCA2 by significantly upregulating E2F-dependent transcription targets and downregulating *CDKN1A* gene expression. *CDKN1A* encodes p21, a CDK inhibitor and PCNAinteracting protein. Whilst BRCA1 or BRCA2 abrogation delays entry into S-phase and slows down replication fork progression, β-catenin activation accelerates both processes, in a p21-dependent manner. Premature S-phase entry induces illegitimate origin firing and fork collapse in BRCA-deficient cells, a subset of which map to actively transcribed genes. Our results suggest that aberrant transcription of specific genes triggered by oncogene activation causes replication failure and cell death in the context of BRCA1/2 deficiency, thereby providing a mechanism for the observed low incidence of oncogene overexpression in tumours with compromised BRCA1/2 function.

## Results

### BRCA1/2 mutations are incompatible with oncogene activation.
Although co-occurrence of *CCND1* and *CCNE1* gene amplification with *BRCA1/2* gene mutations was not found in tumours[18–20], these studies have not indicated whether these amplification events correspond to increased expression of the respective genes. We, therefore, re-analysed The Cancer Genome Atlas (TCGA) data[18] for amplification of known oncogenes in breast ($n = 472$) and ovarian ($n = 311$) cancer patient cohorts, and we investigated whether it correlates with increased mRNA expression. We observed that *BRCA1/2* gene alterations were mutually exclusive with amplification of all the oncogenes examined in the breast (*CCND1*, *CCNE1* and *CDC6*) and ovarian (*CCND1*, *CCNE1* and *BRAF*) cancer cohorts (Supplementary Fig. 1a, b). Importantly, oncogene amplification in these tumours was largely associated with increased mRNA expression of the corresponding oncogenes (Supplementary Fig. 1c, d), indicating that the amplification events correspond to genuine activation of the respective oncogenes. Moreover, we analysed the data from the Memorial Sloan Kettering-integrated mutation profiling of actionable cancer targets (MSK-IMPACT) platform obtained from 1756 breast cancer patients[26], which has higher sequencing depth relative to previous studies (771-fold average coverage), enabling identification of rare, sub-clonal mutations. Here, we determined that mutations in genes of the WNT/β-catenin signalling pathway (*APC*, *AXIN1*, *AXIN2*, *CTNNB1*), known to trigger constitutive, oncogenic β-catenin activation[27–29], were also mutually exclusive with *BRCA1* or *BRCA2* mutations (Supplementary Fig. 1e). Of note, mRNA expression of *CTNNB1* gene encoding β-catenin is likely not affected by mutations in WNT/β-catenin pathway genes, as these stabilise and enhance β-catenin expression by preventing its ubiquitylation and/or proteasomal degradation.

### β-catenin activation is lethal to BRCA1/2-deficient cells.
Next we investigated the effect of inducible oncogene overexpression in cellular models for BRCA1 and BRCA2 inactivation. We focused on β-catenin as its oncogenic expression can be rapidly and reproducibly induced in vitro using chemical inhibitors of glycogen synthase kinase 3 (GSK3). The GSK3β paralogue is an

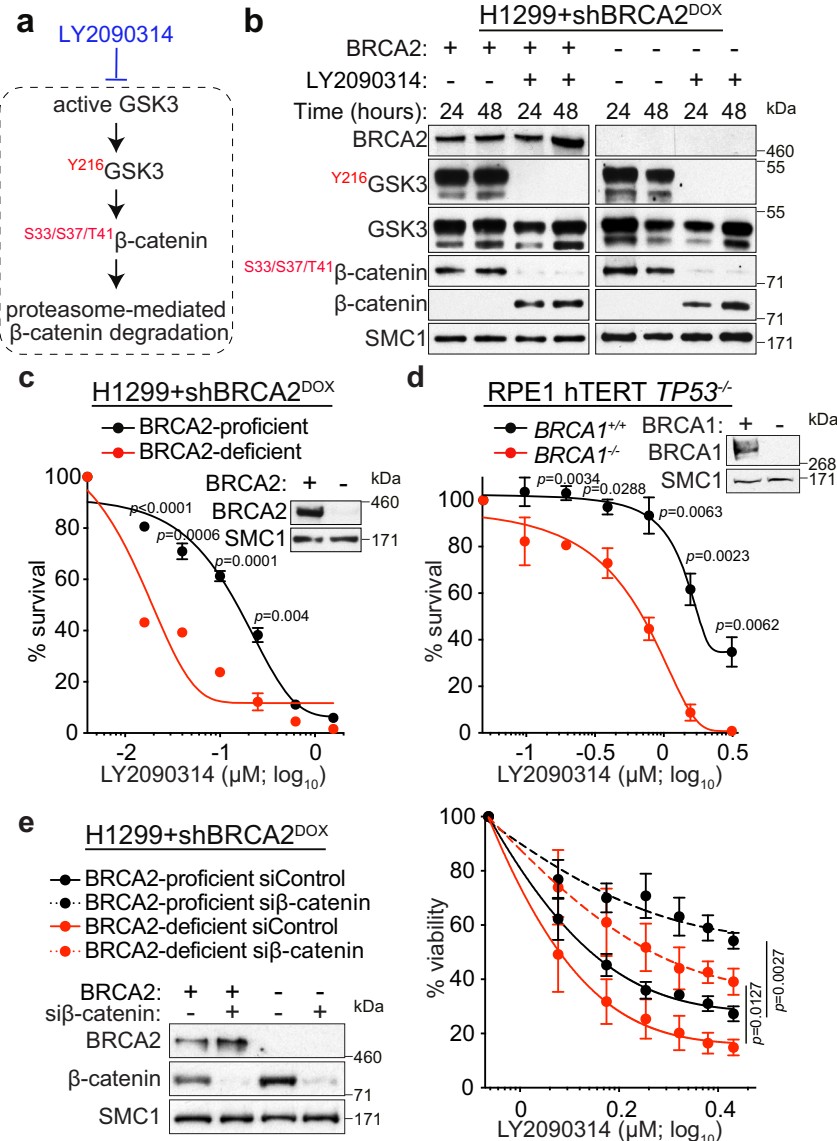

**Fig. 1 GSK3 chemical inhibition triggers β-catenin accumulation, which decreases cell survival in the absence of BRCA1 or BRCA2. a** Diagrammatic representation of GSK3 function in stabilising β-catenin and the effect of GSK3 inhibitor LY2090314. **b** Human H1299 cells carrying a doxycycline (DOX)-inducible BRCA2 shRNA were grown in the presence (−BRCA2) or absence (+BRCA2) of 2 μg/mL DOX, and treated or not with 250 nM LY2090314. Whole-cell extracts prepared at indicated times after LY2090314 addition were immunoblotted as shown. SMC1 was used as a loading control. Phosphorylation sites are indicated in red. Data are representative of $n = 3$ independent experiments. **c, d** Clonogenic survival assays performed in BRCA1/2-proficient or -deficient H1299 (**c**) and RPE1 (**d**) human cells treated with LY2090314 at the indicated doses. Whole-cell extracts prepared at the time of cell seeding were immunoblotted as shown. SMC1 was used as a loading control. Error bars represent standard error of the mean (SEM) of $n = 3$ independent experiments, each performed in triplicate. Statistical significance was determined by an unpaired two-tailed $t$-test. **e** H1299 cells carrying a DOX-inducible BRCA2 shRNA were grown in the presence (−BRCA2) or absence (+BRCA2) of DOX and transfected with control or β-catenin siRNAs. Two days later, cells were treated with LY2090314 at the indicated concentrations for 6 days and processed for cell viability assays. Whole-cell extracts prepared at the start of treatment were immunoblotted as shown. SMC1 was used as a loading control. Error bars represent SEM of $n = 3$ independent experiments, each performed in triplicate. Statistical significance was determined by an unpaired two-tailed $t$-test. Source data for (**b**–**e**) are provided in the Source data file.

enzyme with well-characterised roles in WNT signalling[30]. In this study, we used LY2090314[31], a GSK3 inhibitor (GSK3i) previously shown to stabilise β-catenin by preventing its ubiquitylation and proteasome-dependent degradation (Fig. 1a). As a cellular model for BRCA2 inactivation, we used human H1299 cells carrying a doxycycline (DOX)-inducible shRNA against BRCA2. DOX addition effectively abrogated BRCA2 expression (Fig. 1b). Treatment with 250 nM LY2090314 for 24 or 48 h inhibited both GSK3β autophosphorylation at Tyr216, a marker of active GSK3[32–34], and the GSK3-dependent β-catenin

phosphorylation at Ser33/Ser37/Thr41. Phosphorylation at these sites targets β-catenin for ubiquitylation and degradation, when the WNT pathway is inactive[35]. Ultimately, therefore, GSK3i treatment stabilised β-catenin expression in H1299 cells and induced its accumulation. This was observed in other human cell lines (DLD1, HCT116, and MDA-MB-231; Supplementary Fig. 2a–c), indicating that LY2090314 represents an effective activator of WNT signalling in vitro.

Although β-catenin accumulation was independent of BRCA2 status in all cell lines analysed, clonogenic assays for

cell survival demonstrated that exposure to GSK3i was specifically toxic to H1299 cells lacking BRCA2 (Fig. 1c). Similarly, DLD1 cells with constitutive WNT signalling, driven by an APC mutation[36], showed GSK3i-induced β-catenin accumulation, which in *BRCA2*−/− background was associated with increased sensitivity to GSK3i (Supplementary Fig. 2d). GSK3i toxicity was recapitulated in RPE1 human cells in which *BRCA1* was deleted with CRISPR/Cas9 (Fig. 1d). Thus, GSK3i selectively decreased BRCA1/2-deficient cell survival. To establish whether this effect was mediated by β-catenin, we inhibited its expression using siRNA in BRCA2-proficient and -deficient H1299 cells (Fig. 1e). β-catenin depletion rescued LY2090314 toxicity in H1299 cells lacking BRCA2 and to a lesser extent in BRCA2-proficient cells, suggesting that β-catenin accumulation mediates the effect of GSK3i on cell viability. These results demonstrated a synergistic effect on cell viability between β-catenin activation and BRCA1/2 deficiency, therefore providing a mechanism for the mutual exclusivity observed in breast tumours between gene mutations that promote oncogenic β-catenin activation and *BRCA1/2* gene mutations (Supplementary Fig. 1e).

Our experimental system relies on β-catenin stabilisation and accumulation upon treatment with GSK3i. Therefore, it was essential to test whether the results obtained using chemical GSK3 inhibition can be reproduced with genetic approaches, for example, siRNA-mediated GSK3 depletion. The GSK3 enzyme is an evolutionarily conserved serine/threonine kinase with two paralogs, GSK3α and GSK3β. Although functionally distinct[37], they share 85% overall homology and 98% homology in the catalytic domain[38]. GSK3i inhibit the enzymatic activity of both isoforms[39]. We designed siRNAs that specifically target the GSK3α or GSK3β isoform and depleted these enzymes separately or together in H1299 cells, whilst monitoring cell proliferation (Supplementary Fig. 3a). GSK3α/β co-depletion, but not depletion of each isoform alone, decreased proliferation of BRCA2-deficient H1299 cells, thus recapitulating the effect of LY2090314. Moreover, analyses of TCGA tumour data demonstrated that GSK3α and GSK3β mRNA levels are specifically increased in *BRCA1/2*-mutated breast tumours (Supplementary Fig. 3b), suggesting GSK3 as a potential clinical target for tumours lacking BRCA function.

To investigate whether our observations obtained using oncogenic β-catenin can be extended to other oncogenes, we examined the effect of cyclin D1 and cyclin E overexpression in BRCA1- or BRCA2-depleted human U2OS cells. Firstly, we generated a U2OS cell line with inducible (Tet-ON) cyclin D1 overexpression (Supplementary Fig. 4a). Overexpression of cyclin D1 significantly accelerated S-phase entry after release from mitotic arrest (Supplementary Fig. 4b), confirming its oncogenic activity and consistent with the effect of oncogenic cyclin E overexpression[16]. Cyclin D1 overexpression specifically decreased the proliferation of cells lacking BRCA1 or BRCA2 (Supplementary Fig. 4c), consistent with the mutual exclusivity of *BRCA1/2* mutations and *CCND1* amplification in tumours (Supplementary Fig. 1a, b). Similarly, we observed a significant decrease in the proliferation of BRCA1-depleted U2OS cells in which enhanced cyclin E expression was induced using a Tet-OFF system (Supplementary Fig. 4d).

**β-catenin accumulation alters gene expression**. Oncogenes promote cancer cell proliferation in part through deregulation of specific transcriptional programmes[40]. As a key WNT pathway oncogene[41], β-catenin also controls transcription of multiple target genes[42] via its binding to transcription factors including lymphoid enhancement factor (LEF-1) and T-cell factor (TCF4). Given the established role of β-catenin in transcription, we

hypothesised that the synergy between β-catenin activation and BRCA1/2 deficiency could be triggered by specific β-catenin-dependent alterations in the transcriptome of BRCA-deficient cells. To address this, we performed differential gene expression analyses of RNA-seq data using stringent conditions (FDR < 0.05; |log₂(Fold Change)| > 0.5) to identify genes deregulated by oncogenic β-catenin. As anticipated, in both BRCA2-proficient and -deficient cells we observed prominent upregulation of gene targets of the WNT signalling pathway (Fig. 2a and Supplementary Fig. 5a), including *AXIN2*, *LEF1*, *SOX4* and *TCF7*[43]. Expression of these genes was increased in response to GSK3i treatment, which augmented β-catenin levels and decreased upon β-catenin depletion using siRNA. As a control, we detected downregulation of *CTNNB1* gene encoding β-catenin in cells treated with siRNA against β-catenin (Fig. 2a and Supplementary Fig. 5a).

Overall, β-catenin activation altered the expression of 874 and 1268 genes in BRCA2-proficient and -deficient cells, respectively (Supplementary Fig. 5b), with 517 genes common between the two conditions (Supplementary Fig. 5b). Among these, we identified 241 genes upregulated by β-catenin accumulation in both BRCA2-proficient and -deficient cells (Fig. 2b, blue). In addition to the WNT pathway, upregulated genes belong to estrogen, MYC, KRAS and TNF signalling networks, according to the Molecular Signatures Database (MSigDB) Hallmark gene sets classification (Fig. 2c, blue). We then analysed the 385 genes upregulated by β-catenin specifically in BRCA2-deficient cells. Remarkably, the highest upregulated gene set was that comprising E2F transcriptional targets (Fig. 2c, green). Among these, *CCNE1*, *CCND1* and *CDC6* genes encode cyclin E, cyclin D1 and CDC6 proteins, whose expression increases during G1/S transition to promote entry into S-phase[44]. Moreover, RNA-seq data revealed that β-catenin inhibits *CDKN1A* (encoding p21) expression in BRCA2-deficient cells. p21 is a CDK inhibitor, which sustains G1 arrest under various pathological conditions[45].

To establish whether the alterations in mRNA levels detected by RNA-seq correspond to changes in protein expression, we induced β-catenin accumulation or depleted β-catenin in BRCA2-proficient and -deficient cells and monitored cyclin E, CDC6 and p21 protein levels using immunoblotting. We observed that cyclin E and CDC6 protein levels increased when β-catenin was upregulated, and decreased upon β-catenin depletion in BRCA2-deficient cells (Fig. 2d). Conversely, p21 expression decreased in cells overexpressing β-catenin and was restored when cells were treated with β-catenin siRNA. Moreover, inhibition of p21 expression upon LY2090314-induced β-catenin accumulation was observed in other human cell lines (MDA-MB-231 and HeLa cells; Supplementary Fig. 2e, f). Therefore, the expression of *CCNE1, CDC6* and *CDKN1A* mRNAs correlated with the levels of proteins encoded by these genes. Importantly, using FACS analyses of DNA content we found that β-catenin activation abolished G1 accumulation and promoted S-phase entry in BRCA2-deficient cells, whilst β-catenin depletion restored G1 arrest (Fig. 2e). We reasoned that upregulation of E2F targets and suppression of p21 expression induced by β-catenin could mediate this effect.

**β-catenin accumulation accelerates S-phase entry**. We have previously demonstrated that the G1/S transition is markedly delayed in BRCA2-deficient cells[3]. This is due to chromosome mis-segregation during mitosis which results in DNA damage accumulation in the subsequent G1. To investigate in greater detail how oncogenic β-catenin counteracts the G1/S delay, we synchronised H1299 cells overexpressing β-catenin in mitosis using nocodazole. Following mitotic shake-off, cells were released

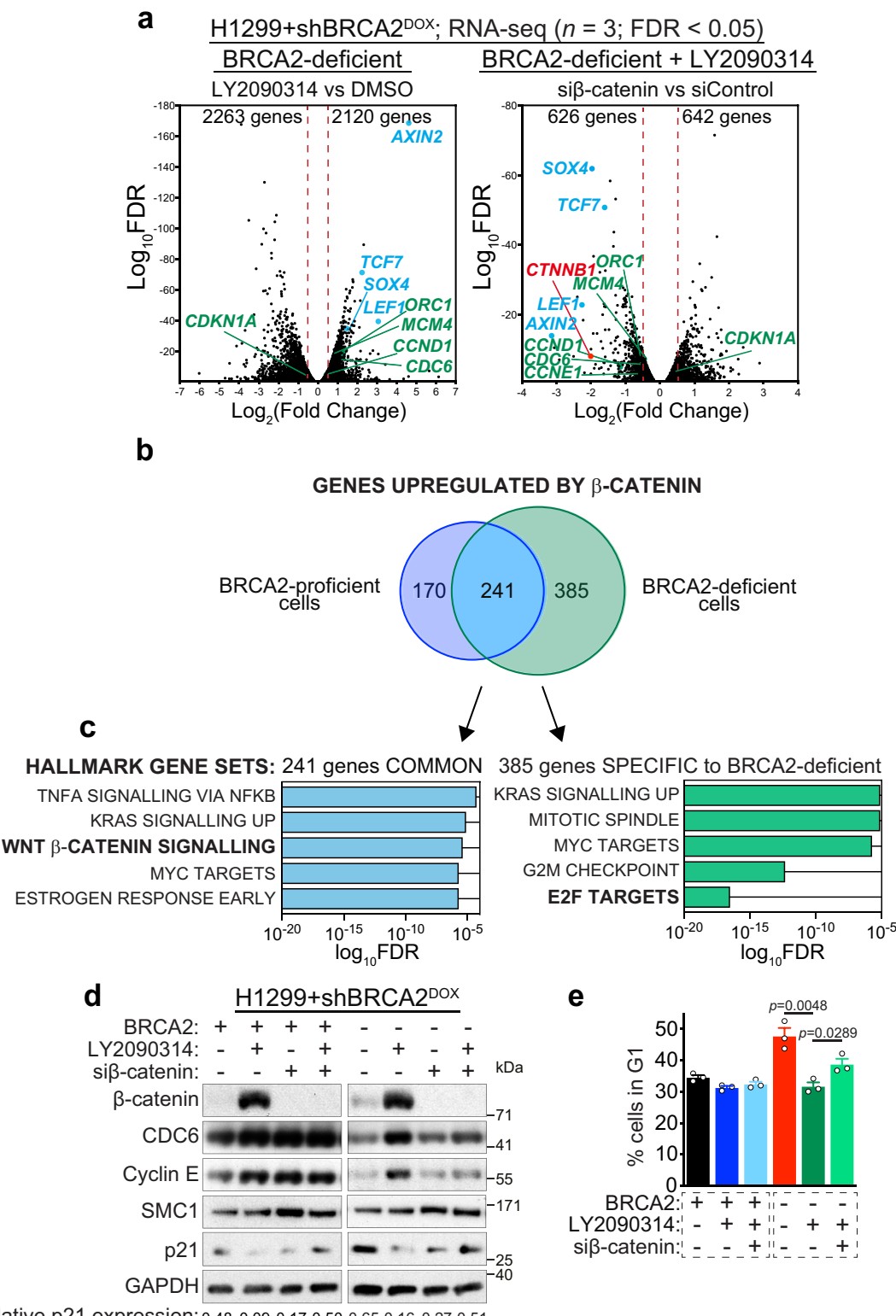

in the presence of the thymidine analogue EdU and allowed to progress synchronously through G1 into S-phase (Supplementary Fig. 6a, b). FACS analyses of EdU incorporation at multiple timepoints after mitotic release enabled us to accurately monitor S-phase entry (Fig. 3a and Supplementary Fig. 6c). BRCA2-depleted cells showed a marked delay in entering S-phase compared to BRCA2-proficient cells, which was effectively reduced by β-catenin accumulation. For example, we detected 18% EdU-positive BRCA2-deficient cells (Fig. 3a, dotted line) 12 h after mitotic shake-off, whilst the percentage increased to 36% EdU-positive cells at the same timepoint, when cells were treated with GSK3i. Moreover, we monitored expression of β-catenin targets cyclin E and CDC6 in cells treated with LY2090314 relative to untreated cells (Fig. 3b). Cyclin E expression was upregulated by GSK3i at the early timepoints after S-phase entry (6 and 8 h), whilst increased CDC6 expression was observed at all timepoints

**Fig. 2 Oncogenic β-catenin deregulates transcription of E2F gene targets and *CDKN1A* in BRCA2-deficient cells. a** BRCA2-deficient H1299 cells transfected with control or β-catenin siRNAs were treated with DMSO or 250 nM LY2090314 for 24 h before processing for RNA-seq (*n* = 3 independent experiments). Volcano plots show differentially expressed genes (FDR < 0.05) in the samples indicated. Gene subsets significantly downregulated (log$_2$(Fold Change) < −0.5) or upregulated (log$_2$(Fold Change) > 0.5) are marked by dotted lines. **b** Venn diagram of genes significantly upregulated by LY2090314 (FDR < 0.05 and log$_2$(Fold change) > 0.5) and downregulated by β-catenin siRNA (FDR < 0.05 and log$_2$(Fold change) < −0.5) in H1299 BRCA2-proficient (purple) and -deficient (green) cells. β-catenin target genes overlapping in the two samples of Venn diagram are indicated in blue (241 genes). **c** Gene set enrichment analysis based on functional annotation (Hallmark gene sets) of β-catenin upregulated genes common between BRCA2-proficient and -deficient cells (blue), or unique to BRCA2-deficient cells (green). **d** H1299 cells expressing a DOX-inducible BRCA2 shRNA were transfected with control or β-catenin siRNAs and grown in the presence (−BRCA2) or absence (+BRCA2) of DOX and LY2090314. Whole-cell extracts were immunoblotted as shown. SMC1 and GAPDH were used as loading controls. p21 expression was quantified relative to GAPDH control. Data are representative of *n* = 3 independent experiments. **e** FACS analyses of DNA content of cells treated as in (**d**), error bars represent SEM of *n* = 3 independent experiments. Statistical significance was determined by an unpaired two-tailed *t*-test. Source data for (**d**, **e**) are provided in the Source data file.

examined, consistent with the GSK3i-induced accelerated S-phase entry observed by FACS. We next evaluated the relative impact of β-catenin accumulation and p21 inhibition on S-phase progression of cells lacking BRCA2 by measuring the percentage of EdU-positive cells 14 h after mitotic shake-off (Fig. 3c). Treatment with siRNA against β-catenin reversed the premature S-phase entry induced by GSK3i, demonstrating that its effect is mediated by β-catenin accumulation. Moreover, p21 inhibition also accelerated G1/S transition in BRCA2-deficient cells and GSK3i treatment did not significantly enhance this effect. This indicates that p21 is a key mediator of the accelerated G1/S transition induced by oncogenic β-catenin.

β-catenin accumulation also enhanced progression into S-phase of *BRCA1*$^{+/+}$ and *BRCA1*$^{-/-}$ RPE1 cells (Supplementary Fig. 7a) and triggered cyclin E expression (Supplementary Fig. 7b). Notably, *BRCA1*$^{-/-}$ cells entered S-phase at remarkably low rates relative to *BRCA1*$^{+/+}$ controls (22% of the *BRCA1*$^{-/-}$ cells were EdU-positive at 12 h post-mitotic release, compared to 10.2 h for *BRCA1*$^{+/+}$ cells; Supplementary Fig. 7a, c dotted line), reflecting a slow recovery from the nocodazole-induced mitotic arrest. Upon β-catenin activation, 40% of *BRCA1*$^{-/-}$ cells became EdU-positive 12 h after mitotic release (Supplementary Fig. 7a, c). Together these results demonstrate that BRCA1- and BRCA2-deficient cells arrest in G1 and delay entry into S-phase, both reversed by β-catenin induction.

We reasoned that accelerated G1/S transition could deregulate replication and activate the intra-S checkpoint. To test this possibility, we induced β-catenin accumulation in BRCA2-proficient and -deficient cells and monitored checkpoint activation using immunoblotting. Surprisingly, CHK1 phosphorylation at Ser317/Ser345, indicative of ATM/ATR activation[46], and RPA32 phosphorylation at Ser33, indicative of single-stranded DNA accumulation, common intermediate at replication-associated lesions[1], were specifically detected in BRCA2-deficient cells with activated β-catenin (Fig. 3d). Cleaved PARP expression, a marker for apoptosis, was also increased in cells lacking BRCA2, supporting the specific toxicity of oncogenic β-catenin against these cells observed in survival assays (Fig. 1c). Oncogene-induced replication defects were proposed to inflict DNA damage in the form of DSBs[13,47]. Consistent with this, we observed an increased frequency of 53BP1 foci in BRCA2-deficient cells upon β-catenin accumulation (Fig. 3e). Thus, β-catenin accumulation in BRCA2-deficient cells triggers premature S-phase entry, checkpoint activation and replication-associated DNA lesions.

**Activated β-catenin triggers origin firing and fork collapse.** Recently, oncogene overexpression was shown to induce illegitimate origin firing events, frequently localised in intragenic regions of the genome[16]. We, therefore, used EdU-seq assays to investigate whether oncogenic β-catenin could deregulate

replication initiation and to determine its consequences when BRCA1 or BRCA2 are abrogated. EdU-seq combines EdU incorporation into nascent DNA with high-throughput sequencing and is established as a high-resolution method for mapping origin firing genome-wide[48]. For EdU-seq, cells overexpressing oncogenic β-catenin were arrested with nocodazole in mitosis and released in the presence of EdU and hydroxyurea (HU). In both BRCA2-deficient H1299 cells and *BRCA1*$^{-/-}$ RPE1 cells, oncogenic β-catenin triggered aberrant origin firing (Fig. 4a, b, red), relative to cells with normal β-catenin expression (Fig. 4a, b, blue). Origin firing at these sites was also less pronounced in BRCA2-proficient H1299 cells and *BRCA1*$^{+/+}$ RPE1 cells upon β-catenin activation (Supplementary Fig. 8a, b). β-catenin-induced origins fired frequently within genes (Fig. 4a, b; red arrows). Overall, we found that the effect of β-catenin upregulation on S-phase entry and activation of novel origins is stronger in BRCA2-deficient cells. We attributed this effect to the fact that cells lacking BRCA2 frequently arrest in G1 and show slower replication fork progression, which enhanced the impact of oncogenic β-catenin, relative to control cells.

We further classified the β-catenin-induced origins based on the intensity of the signal relative to control cells with normal β-catenin expression (Fig. 4c). Constitutive origins were defined as showing 1:1 ratio of EdU signal in GSK3i vs DMSO-treated samples, whilst β-catenin-induced origins showed 2:1 or 4:1 ratio. Using a peak-finding algorithm, we detected in total 4137 and 5109 replication initiating events in BRCA2-deficient cells after 16 and 20 h of release from mitotic arrest, respectively. Among these, 1519 and 1486 origins were induced by β-catenin, corresponding to frequencies of ∼37% and ∼30% at 16 and 20 h, respectively (Fig. 4c and Supplementary Data 1). In BRCA2-proficient cells, ∼23% of the total origins (3496) were induced by β-catenin at 12 h post-release from mitotic arrest (Supplementary Fig. 8c and Supplementary Data 1). Scatter plots of the EdU-seq signal detected in GSK3i- vs DMSO-treated samples further established that there is a clear distinction between the β-catenin-induced and the constitutive origins (Fig. 4d and Supplementary Fig. 8d).

Replication forks initiating at oncogene-induced origins fail to progress and collapse shortly after firing[12,16]. We, therefore, performed experiments in which HU-arrested forks were allowed to restart by HU removal from the media and progressed for 2 h. In contrast to forks initiating at constitutive origins, which recovered and advanced after HU release, forks initiating at β-catenin-induced origins collapsed after release in both BRCA2-proficient and -deficient cells (Fig. 5a and Supplementary Fig. 8e). Thus, oncogenic β-catenin activation triggers aberrant origin firing and forks initiating at these sites fail to progress. Increased RPA phosphorylation was detected only in BRCA2-deficient cells upon treatment with GSK3i (Fig. 3d), suggesting that β-catenin-induced fork stalling/collapse may be more pronounced when BRCA2 is abrogated.

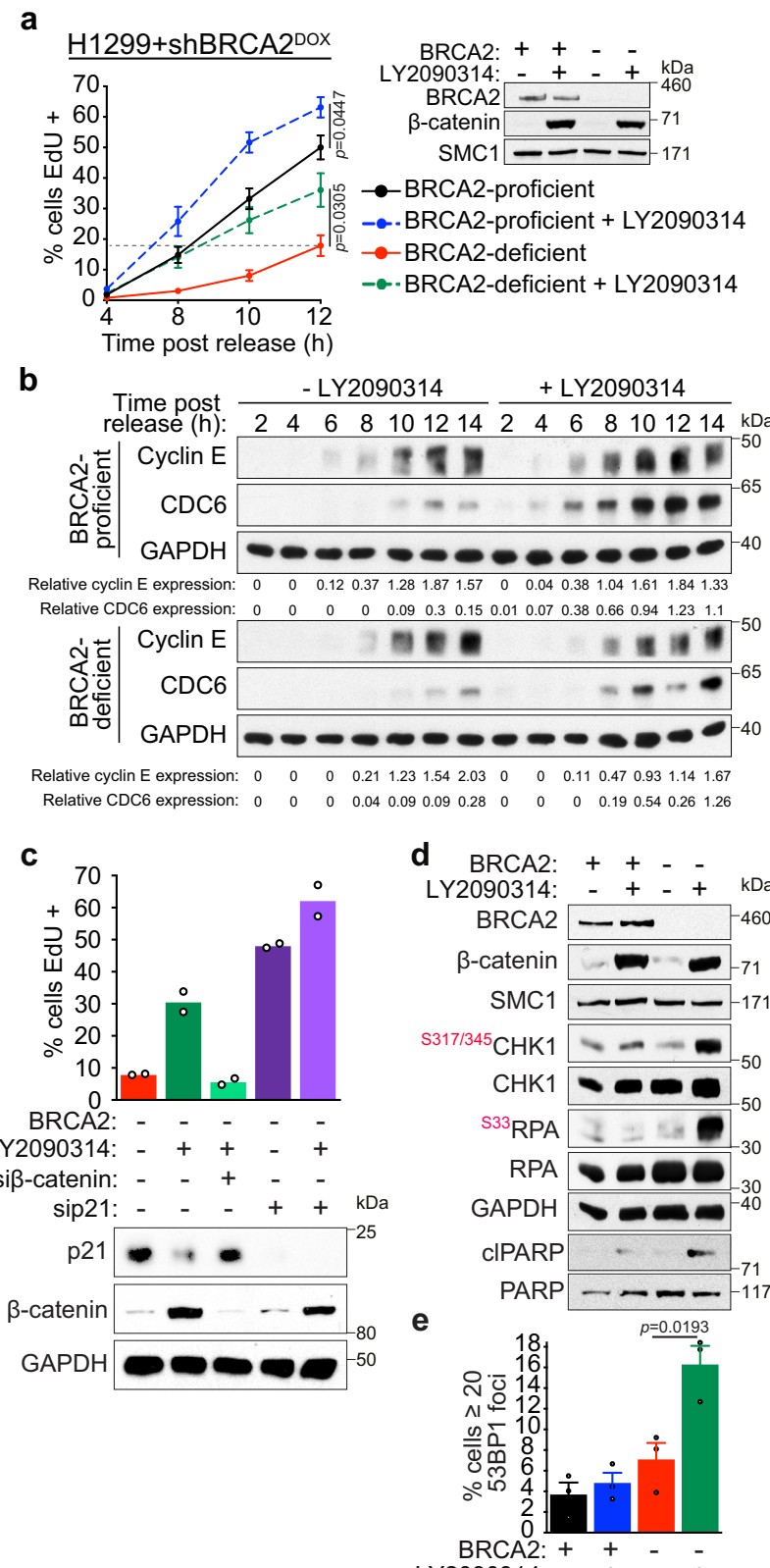

Next, we examined the genome-wide distribution of β-catenin-induced origins with respect to protein-coding genes. In BRCA2-deficient H1299 cells released from mitotic arrest for 20 h, we found that ~52% of the β-catenin-induced origins fired within genic regions, relative to ~59% in BRCA2-proficient cells (Fig. 5b). In *BRCA1*$^{-/-}$ RPE1 cells released for 20 h, the percentage of intragenic origins was ~37%. This was similar to the 41% of intragenic origins triggered by cyclin E over-expression in U2OS cells[16]. Moreover, β-catenin-induced intragenic origin firing was dependent of gene size, with longer genes being more susceptible to illegitimate replication initiation events (Fig. 5c).

**Fig. 3 β-catenin activation accelerates the G1/S transition and triggers the ATR-dependent checkpoint in BRCA2-deficient cells. a** H1299 cells expressing a DOX-inducible BRCA2 shRNA were grown in the presence (-BRCA2) or absence (+BRCA2) of DOX and LY2090314 and arrested in mitosis with nocodazole treatment. Mitotic cells were collected and released in fresh medium containing 25 μM EdU, in the presence or absence of LY2090314. The % of EdU-positive cells was determined using FACS analyses at the indicated time points after mitotic shake-off. Error bars represent SEM of $n = 4$ independent experiments. Statistical significance was determined by an unpaired two-tailed $t$-test. Whole-cell extracts prepared at the time of mitotic shake-off were immunoblotted as shown. SMC1 was used as a loading control. **b** Whole-cell extracts prepared from cells treated as in (**a**) and collected at the indicated time points after release from mitotic arrest were immunoblotted as indicated. GAPDH was used as a loading control. Cyclin E and CDC6 expression were quantified relative to GAPDH control. Data are representative of $n = 2$ independent experiments. **c** BRCA2-deficient H1299 cells were transfected with control, β-catenin or p21 siRNA and grown in the presence or absence of LY2090314 for 24 h. Mitotic cells were collected and released as in (**a**). The % of EdU-positive cells was determined using FACS analyses at 14 h after mitotic shake-off. Mean of $n = 2$ independent experiments. Whole-cell extracts prepared at the time of mitotic shake-off were immunoblotted as shown. **d** BRCA2-proficient or -deficient H1299 cells were grown in the presence or absence of LY2090314 for 24 h and immunoblotted as shown. Phosphorylation sites are indicated in red. Data are representative of $n = 3$ independent experiments. **e** Quantification of frequency of cells with 20 or more 53BP1 foci following treatment as in (**d**). Error bars represent SEM of $n = 3$ independent experiments. Statistical significance was determined by an unpaired two-tailed $t$-test. Source data for (**a**–**e**) are provided in the Source data file.

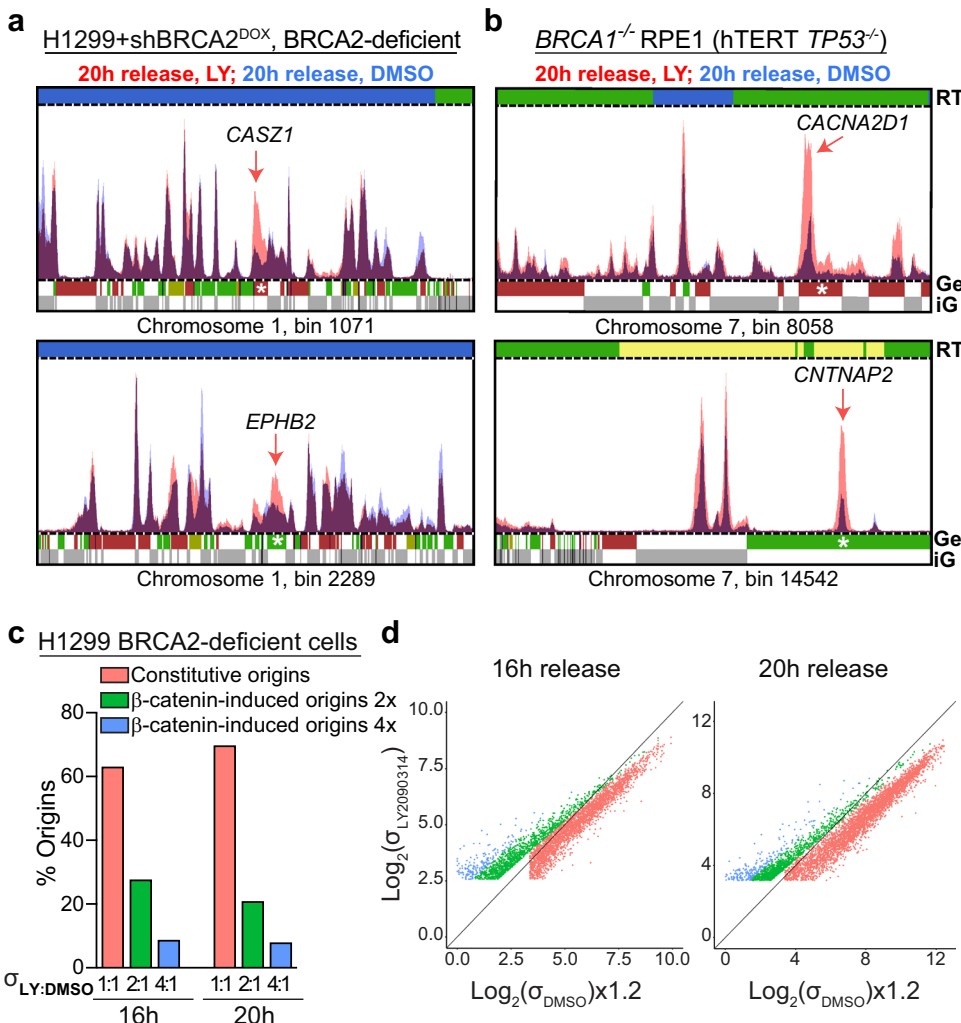

**Fig. 4 β-catenin activation triggers aberrant origin firing in BRCA1/2-deficient human cells. a, b** BRCA2-deficient H1299 (**a**) and $BRCA1^{-/-}$ RPE1 (**b**) cells were grown in the presence (red) or absence (blue) of LY2090314 for 24 h and 100 ng/mL nocodazole was added during the final 8 h of treatment. Mitotic cells were collected and released in fresh medium containing 25 μM EdU and 2 mM HU. EdU-labelled DNA was isolated from cells at 20 h after mitotic release and analysed using high-throughput sequencing. RT, replication timing: blue, early; green, mid; yellow, late S-phase. Ge, genes (green, forward direction of transcription; purple, reverse direction of transcription); iG, intergenic regions (grey). β-catenin-induced origins firing within genes (indicated with white stars) are shown by red arrows. **c** Origin classification based on adjusted σ value ratios in BRCA2-deficient H1299 cells released from mitotic arrest for 16 or 20 h. **d** Scatter plots of EdU-seq σ-values at 16 and 20 h after BRCA2-deficient H1299 cells release from mitotic arrest, colour-coded as in (**c**).

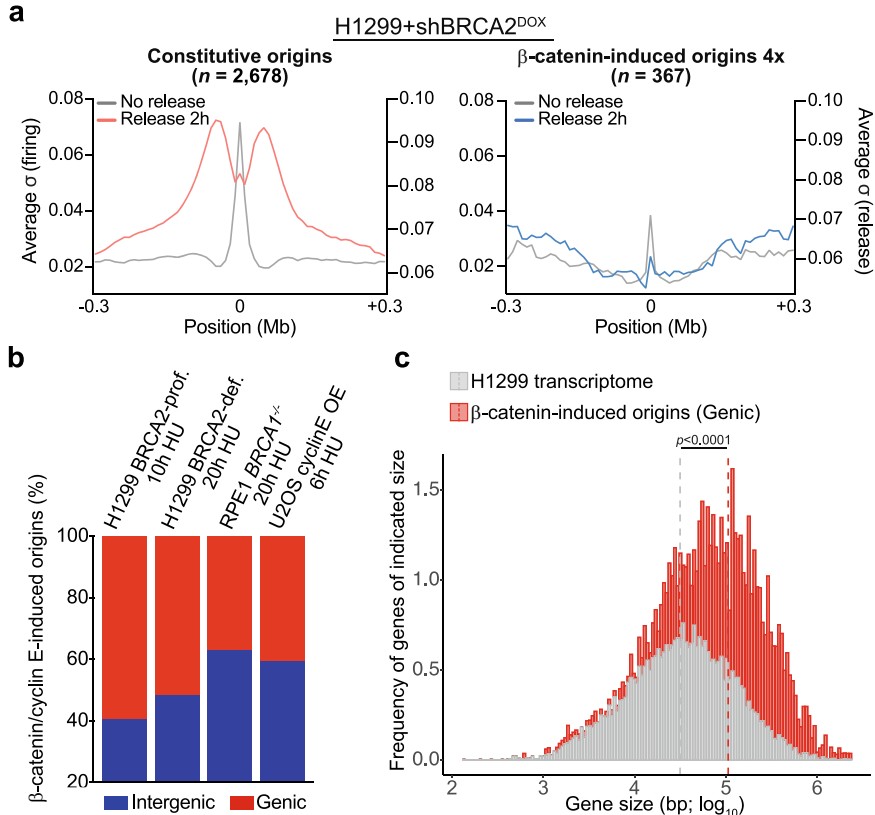

**Fig. 5 β-catenin-induced fork collapse and origin firing in genic vs inter-genic regions. a** Genome-wide average fork progression in BRCA2-deficient H1299 cells treated with 25 μM EdU 30 min before cell collection. **b** Genome-wide distribution of β-catenin-induced or cyclin E-induced origins in BRCA2-proficient and -deficient H1299 cells, BRCA1-deficient RPE1 cells and U2OS cells, based on gene annotation. **c** Size frequency distribution of all protein-encoding genes in H1299 cells (grey) and of protein-encoding genes containing β-catenin-induced origins in BRCA2-deficient H1299 cells (16 and 20 h after mitotic release; red). Dotted lines indicate mean values for each gene set. Statistical significance was determined by a two-tailed Mann–Whitney test. Source data for (**b**) is provided in the Source data file.

**β-catenin activation enhances R-loop formation.** BRCA2-deficient cells have been previously reported to accumulate R-loops likely linked to transcription-replication conflicts[49]. Given that impaired fork progression increases the frequency of transcription-replication conflicts[50–52], we hypothesised that collapsed forks induced by oncogenic β-catenin might enhance R-loop formation in cells lacking BRCA2. As an additional contributing factor to R-loop formation, we also observed a surge in global transcription measured by EU incorporation[53], when β-catenin expression was induced in BRCA2-deficient HeLa cells (Fig. 6a). The β-catenin-dependent transcriptional upregulation was detected in S-phase BRCA2-deficient cells identified using BrdU incorporation (Supplementary Fig. 9a). Moreover, the EU signal was abolished by the inhibitor of transcription elongation 5,6-dichloro-1-β-D-ribofuranosyl-benzimidazole (DRB), confirming that it reflects nascent RNA synthesis (Fig. 6a).

R-loop quantification using S9.6 antibody immunofluorescence demonstrated that BRCA2 abrogation increased R-loop levels compared to control cells, consistent with a previous report[49], and that β-catenin accumulation further enhanced this effect (Fig. 6b). R-loops can be specifically resolved by RNase H1[54]. Therefore, we used HeLa cells expressing a FLAG-tagged, DOX-inducible RNase H1[54] to test whether the S9.6 signal was due to R-loop accumulation (Fig. 6c). In HeLa cells, suppression of p21 expression by GSK3i is dependent on β-catenin (Supplementary Fig. 2f), similarly to H1299 cells. RNase H1 expression significantly reduced the S9.6 signal induced by β-catenin in BRCA2-deficient cells, confirming that it corresponds to DNA–RNA hybrids. Consistently, DNA–RNA immunoprecipitation (DRIP) experiments in which we

analysed R-loop formation at specific loci (*BTBD19*, *APOE* and *RPL13A*), demonstrated that the S9.6 signal is sensitive to RNaseH1 treatment and is enhanced by GSK3i in a β-catenin-dependent manner (Supplementary Fig. 9b–d). To address whether R-loops mediate β-catenin toxicity against BRCA2-deficient cells, we evaluated the impact of RNase H1 expression on the response of these cells to β-catenin activation by GSK3i. Abrogation of R-loops using DOX-inducible RNase H1 expression in HeLa cells increased survival of BRCA2-depleted cells treated with GSK3i (Fig. 6d). These results suggest that R-loop formation is a contributing factor to the sensitivity of BRCA2-deficient cells to oncogenic β-catenin accumulation.

**β-catenin accelerates fork speed by reducing p21 expression.** In addition to G1 arrest, BRCA2 inactivation triggers a significant decrease in replication fork progression[3,55,56]. Given that oncogenic β-catenin abolished the G1 arrest, we investigated whether replication speed was affected in BRCA2-deficient cells, as a consequence of β-catenin-induced premature S-phase entry. Using DNA fibre assays, we found that BRCA2 inactivation reduced replication fork speed relative to control, BRCA2-proficient cells (median speed 0.34 kb/min versus 0.45 kb/min), as previously reported (Fig. 7a). Although β-catenin induction augmented replication rates regardless of BRCA2 status, it had a striking impact in BRCA2-deficient cells, where it increased median replication fork progression from 0.34 kb/min to 0.55 kb/min (~62% increase). Accelerating fork progression by >40% was recently shown to trigger DNA damage accumulation[4,57]. Thus,

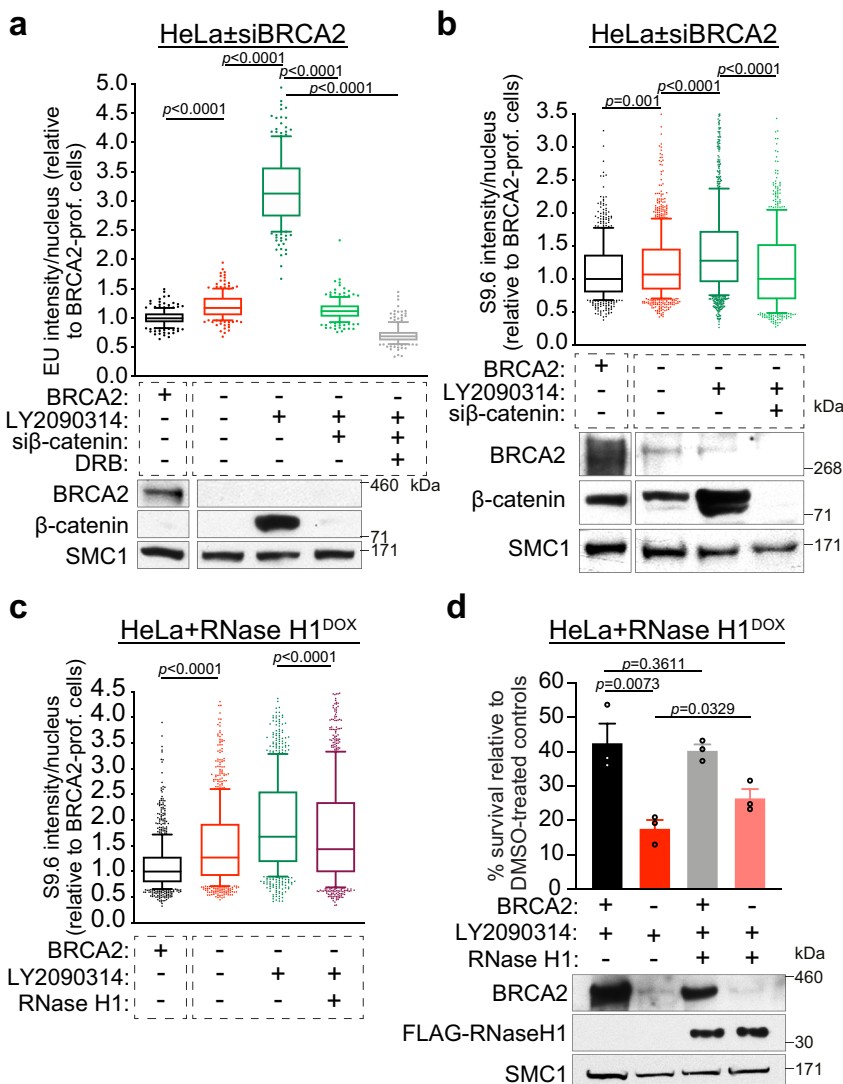

**Fig. 6 Oncogenic β-catenin enhances transcription and R-loop formation in BRCA2-deficient cells. a** HeLa cells transfected with control, BRCA2 or β-catenin siRNAs were grown in the presence or absence of 250 nM LY2090314 for 16 h, with 75 μM DRB added during the final 2 h as indicated, and pulsed with 1 mM EU for 1 h. Whole-cell extracts prepared from each sample were immunoblotted as shown. SMC1 was used as a loading control. EU incorporation was quantified as immunofluorescence intensity per nucleus ($n = 3$ independent experiments; ≥200 nuclei analysed per experiment). Boxes indicate the median value and 25th–75th percentile; whiskers indicate the 10th and 90th percentiles. Statistical significance was determined by a two-tailed Mann–Whitney test. **b** HeLa cells carrying a DOX-inducible FLAG-tagged RNase H1 were transfected with control, BRCA2 or β-catenin siRNAs. Cells were treated with 50 nM LY2090314 for 16 h followed by quantification of S9.6 immunofluorescence per nucleus ($n = 3$ independent experiments; ≥100 nuclei were analysed per experiment). Nucleoli were excluded from the analysis of S9.6 signal. Whole-cell extracts prepared from each sample were immunoblotted as indicated. SMC1 was used as a loading control. Boxes indicate the median value and 25th–75th percentile; whiskers indicate the 10th and 90th percentiles. Statistical significance was determined by a two-tailed Mann–Whitney test. **c** HeLa cells treated as in (**b**) were incubated with 2 μg/mL DOX to induce FLAG-tagged RNase H1 expression. Quantification of S9.6 immunofluorescence was performed as described in (**b**). **d** Clonogenic survival assays were performed in HeLa cells treated as in (**c**) and then incubated with 0.1 μM LY2090314 for 24 h. Survival is expressed relative to DMSO-treated cells. Error bars represent SEM of $n = 3$ independent experiments. Statistical significance was determined by an unpaired one-tailed *t*-test. Whole-cell extracts prepared at the time of cell seeding were immunoblotted as shown. SMC1 was used as a loading control. Source data for (**a–d**) is provided in the Source data file.

abnormally high replication rates caused by β-catenin accumulation in BRCA2-deficient cells are likely to represent a major source of DNA damage and an important contributing factor to the synergistic effect of oncogene activation and BRCA1/2 abrogation on suppressing cancer cell viability.

To investigate the mechanism by which β-catenin accumulation accelerates replication fork progression, we measured the levels of *CDKN1A* gene encoding p21, a modulator of fork speed[4]. Expression of *CDKN1A*, detected by quantitative RT-PCR, was

repressed in cells after β-catenin induction and restored by siRNA-mediated depletion of β-catenin (Fig. 7b). This supported the concept that β-catenin acts as a repressor of *CDKN1A* transcription, which was consistent with our RNA-seq data (Fig. 2a and Supplementary Fig. 5a). To extend the validity of this observation in tumours, we analysed mRNA expression data from the TCGA collections of breast, ovarian, lung and colorectal tumours. Although the number of samples used in these analyses was low, we were nevertheless able to detect an inverse correlation

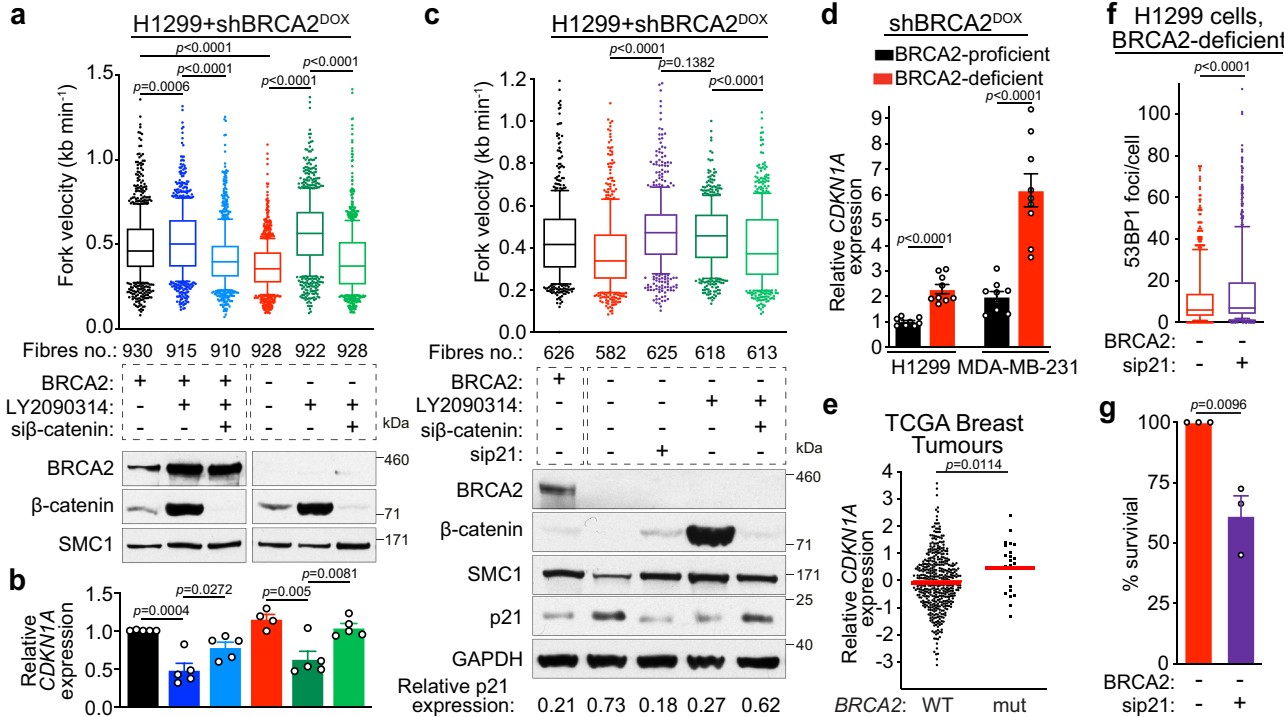

**Fig. 7 Oncogenic β-catenin accelerates replication fork progression by downregulating p21 expression. a** H1299 cells expressing a DOX-inducible BRCA2 shRNA were transfected with control or β-catenin siRNAs and grown in the presence (−BRCA2) or absence (+BRCA2) of DOX, or LY2090314. Cells were pulse labelled with CldU followed by IdU for 20 min each. CldU + IdU track length was used to determine fork velocity. Whole-cell extracts prepared at the time of analysis were immunoblotted with SMC1 as a loading control. Fibres quantified from n = 3 independent experiments. **b** qRT-PCR analyses of H1299 cells grown as in (**a**), for *CDKN1A*. mRNA levels expressed relative to the gene encoding GAPDH and untreated BRCA2-proficient cells $(2^{-\Delta\Delta CT})$. Error bars represent SEM of n = 5 independent experiments. Statistical significance was determined by an unpaired two-tailed *t*-test. **c** DNA fibre analysis as in (**a**) of cells also transfected with p21 siRNA. Whole-cell extracts were immunoblotted with SMC1 and GAPDH as loading controls. p21 expression was quantified relative to GAPDH control. Fibres quantified from n = 2 independent experiments. **d** qRT-PCR analyses as described in (**b**) for H1299 and MDA-MB-231 human cells carrying a DOX-inducible BRCA2 shRNA grown in the presence (BRCA2-deficient) or absence (BRCA2-proficient) of DOX for 28 days. Error bars represent SEM of n = 3 independent experiments. Statistical significance was determined by an unpaired two-tailed *t*-test. **e** *CDKN1A* mRNA expression in TCGA breast tumours (n = 459 for WT; n = 22 for *BRCA2*-mutated;[77]). Each dot represents a single tumour. Middle line, mean. Statistical significance was determined by a two-tailed Mann–Whitney test. **f** Quantification of 53BP1 foci in BRCA2-deficient H1299 cells, 7 days after transfection with control or p21 siRNAs (n = 3 independent experiments; ≥200 cells per experiment). **g** Clonogenic survival assays in BRCA2-deficient H1299 cells transfected with control or p21 siRNAs. Error bars represent SEM of n = 3 independent experiments. Statistical significance was determined by an unpaired two-tailed *t*-test. For **a**, **c** and **f** boxes indicate the median value and 25th–75th percentile, whiskers indicate the 10th and 90th percentiles, and statistical significance was determined by a two-tailed Mann–Whitney test. Source data for (**a**–**d**) and (**f**, **g**) is provided in the Source data file.

between the mean expression of *CTNNB1* gene encoding β-catenin and the mean expression of *CDKN1A* gene encoding p21 (Supplementary Fig. 10a). In other words, mean *CTNNB1* mRNA expression was lower in the tumour subset with high *CDKN1A* expression than in the tumour subset with low *CDKN1A* expression, which was consistent with β-catenin having an inhibitory effect on *CDKN1A* mRNA expression.

β-catenin regulation of *CDKN1A* expression most likely occurs indirectly, via an effector, as neither β-catenin nor its interacting transcription factors, LEF-1 or TCF, are known to regulate the *CDKN1A* promoter. MYC is a well-known inhibitor of *CDKN1A* transcription through direct binding to its promoter[58,59] and MYC expression can be stabilised by β-catenin[60]. Therefore, we tested the hypothesis that β-catenin accumulation leads to increased MYC levels, and this, in turn, inhibits the expression of p21 in our cellular model for inducible BRCA2 inactivation (Supplementary Fig. 10b). β-catenin depletion reduced the expression of MYC protein (Supplementary Fig. 10c), consistent with previously published work[61]. Moreover, siRNA-mediated MYC depletion, as well as MYC chemical inhibition[62,63] increased p21 protein levels in both BRCA2-proficient and

-deficient cells, supporting indirect p21 suppression by β-catenin, through elevated MYC expression (Supplementary Fig. 10d, e).

Interestingly, we detected a higher p21 protein expression in BRCA2-deficient than BRCA2-proficient cells (Fig. 7c), which correlated with reduced rates of fork progression, consistent with the inhibitory effect of p21 on PCNA and DNA synthesis[64,65]. Further supporting this concept, p21 siRNA-mediated depletion augmented replication rates in BRCA2-deficient cells determined using DNA fibre assays for fork progression (Fig. 7c). β-catenin activation caused an increase in fork speed similar to that induced by p21 depletion. This effect is likely due to suppression of p21 expression by β-catenin, because β-catenin siRNA treatment restored low rates of fork progression and augmented p21 expression to the levels similar to untreated BRCA2-deficient cells. Taken together, these results demonstrate that p21 acts to maintain slow replication in cells lacking BRCA2 and that the β-catenin-mediated surge in replication fork progression is due to inhibition of p21 expression in these cells.

In contrast to p21 protein expression which increased after short-term, acute BRCA2 inactivation in human cells (4–6 days; Figs. 2d and 7c), *CDKN1A* mRNA upregulation was most

pronounced after long-term, chronic BRCA2 inhibition in human cells (28 days; Fig. 7d). In addition, immunoblotting of MDA-MB-231 cells indicated that BRCA2-deficient cells had a corresponding increase in p21 protein levels compared to BRCA2-proficient cells, which was suppressed by β-catenin accumulation (Supplementary Fig. 2e). Elevated *CDKN1A* expression was also detected in *BRCA2*-mutated breast tumours relative to those carrying wild type *BRCA2* gene (TCGA data; Fig. 7e). Of note, p53 function is abrogated in both H1299 and MDA-MB-231 cells lines and in the *BRCA2*-mutated TCGA tumours analysed, implicating p53-independent mechanisms in *CDKN1A* activation. It is conceivable that the reduced replication speed caused by p21 upregulation could confer a survival advantage to cells and tumours lacking BRCA2. By enabling activation of alternative pathways for the repair or restart of stalled replication forks, slow replication may prevent the accumulation of replication-associated DNA damage. Consistent with this hypothesis, we found that p21 depletion in BRCA2-deficient cells using two independent siRNAs led to DSB accumulation visualised as 53BP1 foci formation (Fig. 7f, Supplementary Fig. 11a, b). Moreover, loss of p21 reduced survival of BRCA2-deficient cells (Fig. 7g, Supplementary Fig. 11c).

Taken together, these results demonstrate that, by slowing down G1/S transition and limiting replication fork progression, p21 prevents DNA damage accumulation and is, therefore, essential for survival in the absence of BRCA2 (Fig. 8). We find that oncogenic β-catenin inhibits p21 (*CDKN1A*) expression, leading to premature S-phase entry, illegitimate origin firing and accelerated replication fork progression. These caused DNA damage specifically lethal to cells with compromised BRCA1/2 function, which are lacking effective means for fork restoration and DNA repair.

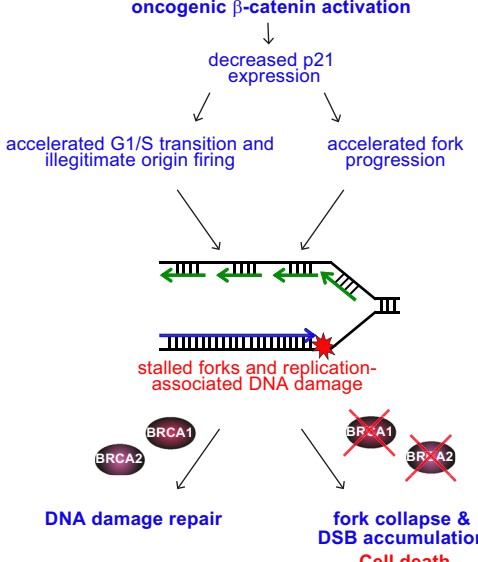

**Fig. 8 Mechanism of action of oncogenic β-catenin in BRCA1/2-proficient and -deficient cells.** Oncogenic β-catenin activation accelerates G1/S transition and fork progression, both mediated by p21 transcriptional downregulation and leading to fork stalling and DSBs accumulation. BRCA1/2-proficient cells can repair oncogene-induced DSBs via homologous recombination. In contrast, in BRCA1/2-deficient cells this homologous recombination repair is abrogated, as well as the mechanisms of fork recovery, leading to DSB accumulation to levels incompatible with cell survival.

## Discussion

Oncogene activation and pathological *BRCA1/2* gene mutations represent well-characterised drivers of tumourigenesis. However, they rarely occur together in cancers. For example, amplification of the *CCND1* gene encoding cyclin D1, a biomarker for poor prognosis in breast cancer[66,67], does not occur in *BRCA1/2*-mutated tumours[20]. Here, we demonstrate that synergistic lethal interactions underpin the mutual exclusivity between oncogene activation and *BRCA1/2* abrogation in cancers, and we identify oncogene-mediated transcriptional deregulation as their driving mechanism.

We focus on β-catenin, an oncogene of the canonical WNT signalling pathway, which has been implicated in tumour initiation, growth and metastasis through transcriptional activation of WNT target genes[21,24]. Whilst ubiquitin-mediated proteolysis maintains β-catenin levels low in normal cells[68], cancer cells often harbour mutations that inactivate the destruction complex consisting of GSK3β, APC and AXIN2, leading to β-catenin stabilisation. Upon translocation into the nucleus, β-catenin associates with members of the TCF/LEF-1 transcription factors family forming a bi-partite transcriptional complex, which alters (represses or activates) the expression of multiple genes[69]. Mutations in the *APC*, *AXIN1/2* and *CTNNB1* genes leading to constitutive WNT signalling through β-catenin activation are frequent in breast and ovarian cancers[27,70]. In breast cancers, oncogenic β-catenin accumulation correlates with cyclin D1 overexpression and both serve as biomarkers for the most aggressive form of the disease[22,71].

Our analysis of published high-throughput sequencing data for breast tumours[26] revealed that mutations in genes of the WNT/β-catenin pathway known to stabilise β-catenin expression and activate its oncogenic potential are mutually exclusive with *BRCA1/2* gene inactivation. This suggested that their co-occurrence is incompatible with cancer cell survival. We tested this hypothesis by inducing β-catenin accumulation in cells lacking BRCA1 or BRCA2, using a chemical inhibitor of GSK3 activity[31]. Accumulation of β-catenin was specifically lethal to cells with compromised BRCA1 or BRCA2 function. This can be extended to other oncogenes, as the same toxicity was recapitulated when cyclin D1 or cyclin E were overexpressed in BRCA1/2-deficient cells. In this context, MYC is an interesting example. In our cellular model for BRCA2-deficiency we find that MYC activation is driven by β-catenin overexpression and leads to p21 downregulation. Therefore, oncogenic MYC is expected to be toxic to cells lacking BRCA2, similarly to other oncogenes tested. Conceivably, overexpression of two oncogenes (β-catenin and MYC) alters transcription and triggers transcription-mediated DNA damage, which is lethal in the context of BRCA2 abrogation. However, these results seem to contradict the report that *MYC* amplification is a driver of *BRCA1*-associated tumorigenesis[72]. According to the mechanism proposed in this study, *MYC* amplification triggers copy number variation resulting in amplifications of other genes, including *MCL1*, which cooperates with *MYC* in sustaining tumour growth. In our cellular model for BRCA2 inactivation, MYC activation driven by β-catenin occurs over a short period of time, which is insufficient for the accumulation of genomic alterations such as pro-survival gene amplifications, explaining why in our experimental conditions MYC overexpression is cytotoxic. Further supporting this concept is our observation that MYC inhibition (using siRNA or a chemical inhibitor) stimulates p21 expression and can therefore promote survival of BRCA2-deficient cells.

To determine the mechanisms underlying β-catenin-induced toxicity in the absence of BRCA2, we examined the transcriptome alterations triggered by oncogenic β-catenin in BRCA2-deficient cells using RNA-seq. This approach enabled us to identify

*CCND1*, *CCNE1*, *CDC6* (all E2F gene targets) and *CDKN1A* encoding p21, as highly deregulated target genes. Changes in mRNA levels correlated with protein levels, as β-catenin activation upregulated cyclin E and CDC6, and inhibited p21 protein expression.

In agreement with the established role of p21 in maintaining G1 arrest under pathological conditions[45], we found that p21 is similarly required in the absence of BRCA1 or BRCA2. G1 arrest in these cells is due to the accumulation of DNA damage in G1, as a consequence of defective mitotic chromosome segregation[3,73]. Consistently, p21 downregulation triggered by oncogenic β-catenin caused premature S-phase entry and aberrant replication origin firing in BRCA1/2-deficient cells. Intragenic replication origins are silenced by transcription during G1. If G1 is shortened, origin silencing fails, leading to replication initiation within actively transcribed genes and to DNA damage[16]. Consistent with this model, we found that oncogenic β-catenin activated a novel set of origins specifically in BRCA1/2-deficient cells, frequently localised within transcribed regions. Notably, origin firing at oncogene-induced origins was also detected in control cells, yet to a lesser extent than upon β-catenin accumulation or cyclin E overexpression[16]. This suggests that transcription-mediated origin inactivation is sporadically incomplete in control cells, leading to illegitimate intragenic origin firing. Forks initiating from these origins failed to progress and collapsed shortly after re-start. Therefore, we propose a two-tier model for transcriptional deregulation caused by β-catenin in BRCA2-deficient cells: first, the direct interaction between β-catenin and TCF/LEF-1 transcription factors alters transcription of specific subsets of target genes, in response to cell-intrinsic signals; second, changes in cell cycle gene expression by β-catenin accelerates G1/S transition and triggers illegitimate origin firing, which inflicts DNA damage.

Oncogenic cyclin E activation was reported to slow down replication by upregulating CDK activity and increasing fork stalling[12,16]. In contrast to cyclin E, we found that oncogenic β-catenin augments the rate of fork progression and that this is critically mediated by inhibition of *CDKN1A* transcription. The protein encoded by this gene, p21, binds PCNA and slows down fork progression[74]. Our results demonstrate that in BRCA2-deficient cells the decrease in p21 levels, as a result of β-catenin activation, speeds up replication fork progression by 62%. This is consistent with the observation that Class IA phosphoinositide 3-kinase (PIK3β) and PKB regulate DNA replication through p21 binding to PCNA[75]. PKB is also known to inactivate GSK3β leading to β-catenin activation. Altered fork progression leads to accumulation of DNA damage, which in the context of BRCA2 deficiency, substantially reduced cell survival. Taken together, these results demonstrate that p21 upregulation provides a mechanism for cell survival upon loss of BRCA2. Conceivably this is reliant on slowing down DNA replication, because this could enable activation of alternative mechanisms for restarting stalled replication forks and repairing replication-associated DNA damage.

We demonstrate that oncogenic β-catenin activation acts synergistically with BRCA1/2 deficiency in eliminating cancer cells and we identify the underlying mechanism as transcriptional deregulation of specific gene targets, with genotoxic consequences. β-catenin acts by reversing two hallmark processes associated with BRCA inactivation: delayed entry into S-phase and low rates of replication fork progression. Strikingly, both accelerated G1/S transition and increased DNA synthesis in BRCA1/2-deficient cells mediated by β-catenin-dependent transcriptional downregulation of *CDKN1A* expression, leading to fork stalling and DSBs accumulation (Fig. 8). As a consequence of stalled forks, β-catenin also exacerbates R-loop formation in BRCA2-deficient cells, which represent an additional source of DSBs. Although β-catenin accumulation stalls replication also in control, BRCA2-proficient cells, these cells can restart forks and repair oncogene-induced DNA damage by homologous recombination. In contrast, BRCA1/2-deficient cells are susceptible to β-catenin accumulation because they lack this major DNA repair pathway and the ability to effectively preserve genome integrity. Consequently, they accumulate DSBs to levels incompatible with cell survival.

Our results provide a mechanism for the mutual exclusivity between oncogenic β-catenin activation and *BRCA1/2* mutations observed in breast tumours. It is conceivable that cells overexpressing β-catenin in conjunction with *BRCA1* or *BRCA2* gene inactivation undergo negative selection during tumourigenesis and die as a consequence of replication-associated DNA damage accumulation. It has been proposed that oncogene activation in premalignant cells causes sufficient DNA damage to induce senescence or cell death thereby preventing tumour onset, unless DNA damage response effectors such as p53 are inactivated[76]. In other words, oncogene activation is tumourigenic only in the absence of functional p53 pathway. Here, we propose that in the context of BRCA1/2 inactivation, oncogene activation is lethal, even when p53 function is abrogated, as is the case in the BRCA1/2-deficient cells and tumours analysed in our study. Thus, oncogene activation has diametrically opposed outcomes in BRCA-proficient and -deficient cancer cells: in the former it provides a mechanism for tumorigenic survival, whilst in the latter it promotes selective elimination, even when DNA damage tolerance is increased by inactivation of key tumour suppressors.

## Methods

**Cell lines and growth conditions.** Human non-small cell lung carcinoma H1299 cells and human invasive ductal breast carcinoma MDA-MB-231 cells carrying a doxycycline (DOX)-inducible BRCA2 shRNA[3], were cultivated in monolayers in Dulbecco's Modified Eagle Medium (DMEM, Sigma-Aldrich) supplemented with 10% tetracycline-free foetal bovine serum (Clontech). For induction of BRCA2 shRNA, 2 μg/mL DOX (Sigma-Aldrich, D9891) was added to growth medium for 3 days. *BRCA2*[+/+] and *BRCA2*[−/−] human colorectal adenocarcinoma DLD1 cells (Horizon Discovery[8]) and human cervical carcinoma HeLa cells carrying a DOX-inducible FLAG-tagged RNase H1 (gift from Prof. Karlene Cimprich, Stanford University, USA[54]) were cultivated in monolayers in DMEM medium supplemented with 10% foetal bovine serum (Sigma-Aldrich). For induction of RNase H1 overexpression, 2 μg/mL DOX was added to growth medium for 2 days. *BRCA1*[+/+] and *BRCA1*[−/−] human retinal pigment epithelial RPE1 cells transduced with hTERT and *TP53*-deleted (a gift from Dr. Dan Durocher, University of Toronto, Canada) were cultivated as above in the presence of 2 μg/mL blasticidin (Life Technologies, A1113903). *BRCA2*[+/+] and *BRCA2*[−/−] human colorectal carcinoma HCT116 cells (Ximbio, Cancer Research Technology) were grown in McCoy's 5a media (Life Technologies) supplemented with 10% foetal bovine serum. Human osteosarcoma U2OS cells carrying a tetracycline-repressible (Tet-OFF) cassette that controls cyclin E expression[16], were cultured in DMEM with 10% foetal bovine serum and 2 μg/mL tetracycline (Tet; Sigma-Aldrich, T7660). Cyclin E overexpression was induced by Tet removal from the media.

To generate human U2OS cells carrying a tetracycline-inducible (Tet-ON) cassette that controls cyclin D1 expression we used the Lenti-X Tet-One Inducible Expression System (Puro; Takara, 631847). Full length *CCND1* cDNA was cloned into the pLVX-TetOne-Puro vector and constructs were introduced into U2OS cells using lentiviral infection. Cyclin D1 Tet-ON U2OS cells isolated from single-cell clones were cultured in monolayers in DMEM supplemented with 10% tetracycline-free foetal bovine serum. For induction of Cyclin D1 expression, 2 μg/mL DOX was added to growth medium for the number of days indicated in figure legend. Unless otherwise indicated, cells were grown at 37 °C in a 5% CO2-containing atmosphere. Cell lines were genotyped and routinely tested for mycoplasma contamination.

The GSK3 inhibitor LY2090314 (Sigma-Aldrich, SML1438-5MG) was added to the media for 24 h at a concentration of 250 nM for H1299 cells and 100 nM for RPE1 cells. H1299 cells treated twice with p21 siRNA were exposed to LY2090314 for 24 or 72 h. For 72 h treatments, all samples were transfected twice. To isolate mitotic cells, H1299, RPE1 or U2OS cells were treated with 100 ng/mL nocodazole (Sigma-Aldrich, M1404) for 8 h. Mitotic cells were collected using mitotic shake-off and subsequently released in fresh growth medium containing 25 μM 5-ethynyl-2′-deoxyuridine (EdU; Invitrogen, C10634) to monitor S-phase entry or 25 μM EdU and 2 mM HU (Sigma Aldrich, H8627) for EdU-seq experiments. To assess fork progression, mitotic shake-off cells were treated with 2 mM HU for the indicated times and then released for 2 h. In the final 30 min 25 μM EdU was added. Cells

treated with the MYC inhibitor were simultaneously treated with 10 μM 10058-F4 (Stratech Scientific, A1169-APE-5mg) and 250 nM LY2090314 for 24 h.

**siRNAs**. Cells were transfected with siRNA using Dharmafect 1 (Dharmacon). Briefly, 0.25–2 × 10^6 cells were transfected with 40 nM siRNA by reverse transfection. Following incubation for 24 or 48 h, depletion was determined by immunoblotting. Sequences of siRNAs used were as follows: siβ-catenin 5′-AUC AAC UGG AUA GUC AGC ACC-3′ (Dharmacon), siBRCA1 5′-CAG CAG UUU AUU ACU CAC UAA-3′ (Qiagen), siGENOME BRCA2 SMARTpool (Dharmacon, M-003462-01), siCDKN1A (p21, Dharmacon, J-003471-11), siCDKN1A (p21, Ambion, s416), siGSK3α (Thermo Fisher Scientific, 4390824; s6238), siGSK3β (Thermo Fisher Scientific, 4390824; s6241), siGSK3α/β (Cell Signaling, 6301), siMYC SMARTpool (Dharmacon, L-003282-02). AllStars siRNA (Qiagen, 1027281) was used as a negative control.

**Clonogenic survival assays**. Cells were plated in triplicate at densities between 200 and 600 cells per well in 6-well plates. H1299 cells (3 days after DOX treatment) and RPE1 cells were treated a day after seeding with LY2090314 at the indicated doses for 24 h, followed by replacement of fresh media. RPE1 cells were cultured at 37 °C in a 5% CO$_2$ and 3% O$_2$-containing atmosphere. HeLa cells were treated with DOX to induce expression of FLAG-tagged RNase H1 and transfected with control or BRCA2 siRNAs 2 days before seeding. BRCA2-deficient H1299 cells were transfected twice with p21 siRNA before seeding. Colonies were stained with 5 mg/mL crystal violet (Sigma-Aldrich, 1159400025) in 50% methanol and 20% ethanol. Cell survival was expressed relative to control cells.

**Cell proliferation and viability assays**. H1299 cells were cultured in the presence or absence of 2 μg/mL DOX for 3 days. Three days after the addition of DOX cells were transfected with control siRNA or siRNA against β-catenin, GSK3α and/or GSK3β. Two days later, cells were seeded in triplicate in 96-well plates.

U2OS cells carrying a tetracycline-repressible (Tet-OFF) cassette for inducible cyclin E expression were cultured in media supplemented (endogenous cyclin E expression) or not (cyclin E overexpression) with 2 μg/mL DOX. Eight days after the removal of DOX (TET) the cells were transfected with control or BRCA1 siRNA and two days later seeded in triplicate in 96-well plates. U2OS cells carrying a tetracycline-inducible (Tet-ON) cassette for inducible cyclin D1 expression were cultured in media supplemented (cyclin D1 overexpression) or not (endogenous cyclin D1 expression) with 2 μg/mL DOX. Eight days after DOX addition, cells were transfected with control, BRCA1 or BRCA2 siRNAs and two days later seeded in triplicate in 96-well plates.

Viability was determined using resazurin-based assays at the indicated time points. Cells were incubated with 10 μg/mL resazurin (Sigma-Aldrich, R7017) in growth medium at 37 °C for 2 h. Fluorescence was measured (544 nm excitation and 590 nm emission) using POLARstar Omega software and plate reader with data exported using Omega Data Analysis software (BMG Labtech).

**RNA-seq analysis**. H1299 cells were cultured in presence or absence of DOX for 3 days. A day after the addition of DOX both cultures were transfected with control siRNA or siβ-catenin. The following day 250 nM LY2090314 was added to the media for 24 h. Cells were then collected for RNA extraction using the RNeasy Mini Kit (Qiagen, 74104) according to the manufacturer's guidelines. RNA samples were quantified using RiboGreen (Invitrogen) on the FLUOstar OPTIMA plate reader (BMG Labtech) and RNA size profile and integrity were analysed on the 2200 or 4200 TapeStation (Agilent, RNA ScreenTape). RNA integrity number estimates for all samples were between 9.0 and 10.0. Poly(A) transcript enrichment and strand specific library preparation were performed using TruSeq Stranded mRNA kit (Illumina) following the manufacturer's instructions. Libraries were amplified (15 cycles) on a Tetrad (Bio-Rad) using in-house unique dual indexing primers. Individual libraries were normalised using Qubit and size profile was analysed on the 2200 or 4200 TapeStation. Individual libraries were pooled together and diluted to ~10 nM for storage. Each library aliquot was denatured and further diluted prior to loading on the sequencer. Paired-end sequencing was performed using a HiSeq4000 75 bp platform (Illumina, HiSeq 3000/4000 PE Cluster Kit and 150 cycle SBS Kit), generating a raw read count of >22 million reads per sample.

**RNA-seq data processing**. Reads were aligned to the human reference genome (GRCh37) using HISAT2 and duplicate reads removed using the Picard MarkDuplicates tool (Broad Institute). Reads mapping uniquely to Ensembl-annotated genes were summarised using featureCounts. The raw gene count matrix was imported into the R/BioConductor environment for further processing and analysis. Genes with low read counts (less than ~10 reads in more than 3 samples) were filtered out, leaving sets of 17,800–18,500 genes to test for differential gene expression.

**Differential gene expression analysis**. The analyses were carried out using the R package DESeq2 version 1.26.0. Unless otherwise stated, differentially expressed genes were identified based on two criteria: FDR (False Discovery Rate using

Benjamini–Hochberg adjusted $p$-values) <0.05 and absolute value of log$_2$(Fold Change) >0.5. Differentially expressed genes were determined independently for the following treatments of H1299 cells: +DOX vs −DOX, −DOX + LY2090314 vs −DOX, -DOX + LY2090314 + siβ-catenin vs −DOX + LY2090314, +DOX + LY2090314 vs +DOX, and +DOX + LY2090314 + siβ-catenin vs +DOX + LY2090314.

**Gene set enrichment analysis**. The pathway enrichment analyses were performed using the Gene Set Enrichment Analysis (GSEA) software with the Molecular Signatures Database collection (Hallmark gene sets).

**TCGA data set analysis**. The Cancer Genome Atlas (TCGA) breast cancer annotated mutation files for 481 samples were retrieved from the cBioPortal database (http://www.cbioportal.org/). mRNA expression of GSK3α or GSK3β was evaluated in samples with and without BRCA1/2 somatic mutations. Expression of CDKN1A mRNA was determined in samples with and without BRCA2 somatic mutations.

Expression levels (mRNA) and annotated alterations for six genes (BRCA1, BRCA2, BRAF, CCND1, CCNE1 and CDC6) were retrieved from the cBioPortal database (http://www.cbioportal.org/) for Ovarian Serous Cystadenocarcinoma patients (311 samples[18]) and Breast Invasive Carcinoma patients (472 samples[77]). Mutual exclusivity between mutations in the BRCA1/2 genes and amplifications in the BRAF, CCND1, CCNE1 and CDC6 oncogenes was tested using the R implementation of the DISCOVER algorithm. Similarly, mutual exclusivity was tested between mutations in BRCA1/2 genes and APC, AXIN1, AXIN2 and CTNNB1 WNT pathway genes in Breast cancer patients (1,756 samples[26]).

CTNNB1 expression was determined in TCGA data sets of Breast Invasive Carcinoma[77], Ovarian Serous Cystadenocarcinoma[18], Lung Squamous Cell Carcinoma (PanCancer Atlas) and Colorectal Adenocarcinoma (PanCancer Atlas) patients with high (relative mRNA expression >2, except for lung cancer >1.5) or low (relative mRNA expression < −2, except for lung cancer < −1.5) CDKN1A mRNA expression.

**Quantitative RT-PCR**. For RNA reverse transcription, the Ambion *Power* SYBR Green Cells-to-CT Kit (ThermoFisher, 4402954) was used according to the manufacturer's instructions. The resulting complementary DNA was analysed using SYBR Green technology in the Applied Biosystems StepOnePlus Real-Time PCR System with StepOne Software using primers for CDKN1A and GAPDH listed in Supplementary Table 1. After normalisation to GAPDH, gene expression was calculated relative to untreated control cells, as $2^{-\Delta\Delta CT}$.

**Immunofluorescence**. Cells grown on coverslips were washed in PBS then swollen in hypotonic solution (85.5 mM NaCl and 5 mM MgCl$_2$ pH 7) for 5 min. 4% paraformaldehyde fixation was performed for 10 min at room temperature and then cells were permeabilised with 0.5% Triton X-100 (Sigma-Aldrich, X100) in PBS. After blocking in antibody diluting buffer (ADB; 1% goat serum, 0.3% BSA, 0.005% Triton X-100 in PBS) for 30 min, cells were incubated at room temperature with primary antibody diluted in ADB for 2 h. After three washes (ADB/0.4% Photo-Flo, ADB/0.005% Triton X-100, dH$_2$O/0.4% Photo-Flo [Kodak, 1464510]) coverslips were incubated for 1 h at room temperature with fluorochrome-conjugated secondary antibodies (1:1000, Molecular Probes, Invitrogen). Coverslips were washed twice for 5 min (ADB/0.4% Photoflo then 0.005% Triton X-100 in PBS) then dried and mounted on microscope slides using the ProLong Gold Antifade Mountant with 40,60-diamidino-2-phenylindole (DAPI; Life Technologies, P36935). Samples were viewed with a Leica DMI6000B inverted microscope and fluorescence imaging workstation equipped with HCX PL APO ×100/1.4–0.7 oil objective. Images were analysed using ImageJ software (National Healthcare Institute, USA).

**Immunoblotting**. To prepare whole-cell extracts, cells were washed once in 1× PBS, harvested by trypsinisation, washed in 1× PBS and re-suspended in SDS-PAGE loading buffer (0.16 M Tris-HCl pH 6.8, 4% SDS, 20% glycerol, 0.01% bromophenol blue, 100 mM DTT). Samples were sonicated and heated at 70 °C for 10 min. Equal amounts of protein (20–80 μg) were analysed by gel electrophoresis followed by Western blotting. NuPAGE-Novex 10% Bis-Tris (MOPS buffer, Life Technologies) and NuPAGE-Novex 3–8% Tris-Acetate gels (Life Technologies) were run according to the manufacturer's instructions. Proteins were transferred to 0.2 μM nitrocellulose membranes (GE Healthcare) using semi-dry transfer in transfer buffer (Life Technologies, NP00061) with 10% methanol. Membranes were blocked in 5% skimmed milk in PBS-Tween (PBST; 0.05% Tween 20 [Sigma-Aldrich, P7949] in PBS) then incubated overnight at 4 °C in primary antibody diluted in 2% bovine serum albumin (BSA; Sigma-Aldrich, A7906) in PBST. After washes with PBST, the membranes were incubated with anti-mouse or anti-rabbit horseradish peroxidase (HRP)-conjugated secondary antibodies (1:5000, DAKO, P0447 and P0448 respectively) at room temperature for 1 h. The membranes were washed with PBST and developed by enhanced chemiluminescence (ECL; GE and Millipore).

**EdU incorporation in cells**. Cells were synchronised in mitosis, collected and released in fresh growth medium containing 25 µM EdU. Subsequently, samples were collected every 2 h for 14 h by trypsinisation and fixed using 90% methanol. Cell cycle distribution of unsynchronised cells was analysed by incorporating 10 µM EdU for 45 min. Incorporated EdU was detected using the Click-iT EdU Alexa Fluor 647 Flow Cytometry Assay Kit (Invitrogen, C10634) according to the manufacturer's instructions. Cells were re-suspended in 1× PBS containing 20 µg/mL propidium iodide (Sigma-Aldrich, P4864) and 400 µg/mL RNase A (Invitrogen, 12091-021). Samples were processed using flow cytometry (FACSCalibur and BD CellQuest Pro, BD Biosciences). Between 5000 and 10,000 events were analysed per condition using FlowJo software.

**EdU-seq data processing**. After EdU incorporation, samples were processed for EdU-seq[16,48]. Sequence reads were aligned to the masked human genome assembly (GRCh37/hg19) using the Burrows–Wheeler aligner[78], retaining only the reads with base quality scores ≥60. Briefly, each chromosome was split into 10 kb bins and the number of reads for each bin was determined. Next, sigma (σ) values for each genomic bin were calculated, as the normalised number of sequence reads per bin, divided by its standard deviation. The σ values thus obtained were used to identify replication origins, by searching for local maxima (peaks) in each chromosome. If the ratio of the LY2090314:DMSO sigma values was greater than 4, then the origin was considered as LY2090314-induced 4-fold (4:1); similarly, if the ratio was greater than 2, but lower than 4, the origin was considered as LY2090314-induced 2-fold (2:1). All other origins were considered to be constitutive. For EdUseq, we performed at least two independent experiments for each genotype under similar conditions.

**Assignment of genic and intergenic origins**. Ensembl gene annotations (v99 for GRCh37/hg19) were used to create a list of all human genes and their position in the genome. Genomic bins were defined as genic, if they mapped entirely within genes; intergenic, if they mapped entirely within intergenic sequences. If they contained both genic and intergenic sequences, they were classified as mixed genic/intergenic. The analysis of the distribution of origins in the genome considered the intergenic and mixed genic/intergenic bins as intergenic, and the genic bins regardless of transcription direction as genic.

**DNA fibre assay**. Newly replicated DNA was labelled by the addition of 25 µM 5-chloro-2′-deoxyuridine (CldU; Sigma-Aldrich, C6891) to the media, followed by 20 min incubation at 37 °C. Next, cells were washed three times with warm 1× PBS and fresh media containing 250 µM 5-iodo-2′-deoxyuridine (IdU; Sigma-Aldrich, I7125) was added to each well. After incubation for 20 min at 37 °C, cells were harvested by trypsinisation and $5 \times 10^5$ cells were re-suspended in cold 1x PBS. Next, 7 µL of lysis buffer (200 mM Tris-HCl pH 7.4, 50 mM EDTA [Sigma-Aldrich, E9884], 0.5% SDS) were mixed with 2 µL of cell suspension on a microscopy slide and incubated horizontally for 5 min at room temperature. The DNA was spread by tilting the slide manually at an angle of 30–45°. The air-dried DNA was fixed in methanol/acetic acid (3:1) for 10 min. Slides were rehydrated in 1× PBS twice for 5 min and the DNA was denatured in 2.5 M HCl (VWR, 20252.244) for 1 h at room temperature. The slides were washed several times in 1× PBS until a pH of 7–7.5 was reached. Slides were incubated in blocking solution (2% BSA, 0.1% Tween 20, PBS; 0.22 µM filtered) for 40 min at room temperature and in primary antibodies (rat anti-CldU, 1:500, Abcam, ab6326 and mouse anti-IdU, 1:100, Becton Dickinson, 347580) overnight at 4 °C. After five washes in PBS-Tween (0.2% Tween 20 in PBS) for 3 min and one short wash in blocking solution, the slides were incubated with the secondary antibodies (anti-rat Alexa Fluor 555 and anti-mouse Alexa Fluor 488, 1:300; Invitrogen, A21434 and A11001, respectively) for 1 h at room temperature. Subsequently, they were washed as before, air-dried and mounted in ProLong Gold Antifade Mountant (Thermo Fisher Scientific, P36930). Images were acquired from at least 2 independent experiments as described for immunofluorescence and analysed using ImageJ software. Fork velocity was calculated using a conversion factor of 1 µM = 2.59 kb/min[79].

**EU immunofluorescence**. Cells on coverslips were cultured in the absence or presence of LY2090314. As a control, a set of samples were treated with 75 µM DRB (Sigma-Aldrich, D1916) for 2 h. Newly synthesised RNA transcripts were labelled with 1 mM 5-ethynyl uridine (EU; Invitrogen, C10328) for 1 h. Where specified, cells were also labelled with 1 µM BrdU for 1 h. Incorporated EU was detected using the Click-iT RNA Alexa Fluor 594 HCS Assay kit (Invitrogen, C10328) according to the manufacturer's instructions. For samples labelled with BrdU the coverslips were denatured with 2 M HCl for 30 min prior to the Click-IT reaction. Coverslips were then stained for BrdU according to the immunofluorescence protocol (secondary antibody 1:200). The coverslips were air-dried and mounted in ProLong Gold Antifade Mountant. Images were acquired as described for immunofluorescence and analysed using ImageJ software.

**S9.6 immunofluorescence**. For visualisation of DNA–RNA hybrids, S9.6 (hybridoma cell line HB-8730) and nucleolin immunofluorescence was performed[80]. Cells were treated with a pre-extraction solution (0.5% Triton X-100, 20 mM HEPES-KOH, 50 mM NaCl, 3 mM MgCl₂, and 300 mM sucrose) before methanol fixation. Subsequently, coverslips were incubated with anti-DNA–RNA hybrid (S9.6) and anti-nucleolin antibody (Abcam, ab50279; both 1:1000 in blocking solution) diluted in blocking buffer (2% BSA in PBS) overnight at 4 °C. After three washes with 1× PBS coverslips were incubated with secondary antibodies (anti-rabbit Alexa Fluor 488 and anti-mouse Alexa Fluor 594, 1:1000, Invitrogen, A11008 and A11005, respectively) for 1 h at room temperature. After washing as before, nuclei were counterstained with DAPI, coverslips were dried and mounted. Images were acquired using a Leica DM6000 microscope equipped with a DFC390 camera (Leica) at ×63 magnification, all the fields imaged were randomly chosen in the DAPI channel. Automatised quantitation was performed using the Metamorph v7.5.1.0 software (Molecular Probes), including the final analysis where the S9.6 signal in the nucleoli was subtracted from the integrated nuclear S9.6 signal.

**DNA–RNA immunoprecipitation (DRIP)**. DNA–RNA hybrids were immunoprecipitated[80] from extracted and enzymatically digested DNA (treated or not with RNase H) using the S9.6 antibody. Quantitative PCR was performed using primers for BTBD19, APOE, and RPL13A listed in Supplementary Table 1. The relative abundance of DNA–RNA hybrids was normalised to input values and to the control siRNA.

**Antibodies**. The following antibodies were used for immunoblotting: mouse monoclonal antibodies raised against BRCA1 (1:1000, Calbiochem, OP92), BRCA2 (1:1000, Calbiochem, OP95), CDC6 (1:1000, Santa Cruz, sc-9964), CHK1 [G-4] (1:1,00, Santa Cruz, SC-8408), Cyclin E [HE12] (1:1000, Santa Cruz, sc-247), FLAG [M2] (1:5,000, Stratagene, 200472-21), GAPDH [6C5] (1:30,000, Novus Biologicals, NB600-502), RPA [9H8] (1:500, Abcam, ab2175); rabbit polyclonal antibodies raised against β-catenin (1:1000, Cell Signaling, 9562S), cyclin D1 (1:1000, Cell Signaling, 2922S), phosphorylated Ser33/Ser34/Thr41 β-catenin (1:500, Cell Signaling, 9561), phosphorylated Ser317 CHK1 (1:2000, Cell Signaling, 2344), phosphorylated Ser345 CHK1 (1:1000, Cell Signaling, 2341), GSK3α/β (1:1000, Cell Signaling, 5676S), phosphorylated Y279/Y216 GSK3α/β (1:1000, Abcam, AB75745), cleaved PARP [Asp214] [D64E10] (1:1000, Cell Signaling, 5625S), PARP [46D11] (1:1000, Cell Signaling, 9532S), phosphorylated Ser33 RPA (1:1000, Bethyl Laboratories, A300-246A-1), SMC1 [BL308] (1:10,000, Bethyl Laboratories, A300-055A), p21 Waf1/Cip1 [12D1] (1:1000, Cell Signalling, 2947S). Immunofluorescence was performed with the mouse monoclonal antibody: BrdU (1:100, Becton Dickinson, 347580), and the rabbit polyclonal antibody: 53BP1 (1:5000, Novus, NB100-304), nucleolin (1:1000, Abcam, ab50279). Monoclonal S9.6 antibody (1:1000, hybridoma cell line HB-8730) was used for immunofluorescence and DRIP analysis.

**Statistical analysis**. Data were collected and processed in Microsoft Excel and analysed in GraphPad Prism 9. Data were compared using a one-sided or two-sided unpaired t-test or two-tailed Mann–Whitney test and mutual exclusivity was tested by pairwise DISCOVER method.

**Reporting summary**. Further information on research design is available in the Nature Research Reporting Summary linked to this article.

## Data availability
Sequencing data are accessible at the Gene Expression Omnibus (GEO) repository, under accession number GSE153736. The TCGA and MSK data are available in the cBioPortal for Cancer Genomics database (https://www.cbioportal.org). All data is available from the authors upon reasonable request. Source data are provided with this paper.

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

## Acknowledgements

We are grateful to members of M.T. laboratory for critical reading of the manuscript and valuable feedback. Andrés Aguilera's laboratory is funded by the European Research Council (grant ERC2014 AdG669898 TARLOOP). The work in Thanos Halazonetis' laboratory was supported by grants from the Swiss National Science Foundation (182487) and the European Commission (ERC project REPLISTRESS). Research in Madalena Tarsounas' laboratory is supported by Cancer Research UK and University of Oxford. This project has received funding from long-term EMBO non-stipendiary fellowship (ALTF 1044-2019), the European Union's Horizon 2020 research and innovation programme under the Marie Skłodowska-Curie grant agreement No. 722729 and grant agreement No. 886045. M.T. was the recipient of a Mayent-Rothschild-Institute Curie Award.

## Author contributions

M.T. designed the study. R.A.D., G.Z., G.G.R., F.J.G., S.B. and E.M.C.T. performed the experiments and acquired the data. G.G.R. and T.D.H. processed the EdUseq data. B.W. and H.L. processed the RNA-seq data. E.P.L. analysed genomic data. S.B. and A.A. performed and analysed R-loop and DRIP experiments. M.T. and R.A.D. wrote the manuscript. All authors contributed to the interpretation of the results and commented on the manuscript.

## Competing interests

The authors declare no competing interests.
