## [Peer Review File · Nature Communications]

REVIEWER COMMENTS

Reviewer #1 (Remarks to the Author):

Mutations in BRCA1/2 genes are associated with various human cancers affecting breast, ovarian, prostate and other tissues. BRCA1/2 handles the replication stress by stabilising stalled replication forks and conducting homologous recombination repair (HRR) permitting cell cycle progression. A deficiency of BRCA1/2 causes a collapse of the stalled replication forks thereby promoting cytotoxic double strand breaks (DSBs), which ultimately lead to genomic instability and the development of cancer. Oncogene overexpression accelerates the proliferation of (cancer) cells, namely elevated replication stress. Therefore, the coordination of the progression of replication in combination with a proficient repair of the stalled replication forks, is key to prevent genomic instability, cell cycle block and cell death. In case of BRCA1/2 deficiency, oncogene overexpression is even more cytotoxic to cells, which is likely due to the clash of unprogrammed replication firing and a deficiency of DSBs. This approach, thus, may be considered as a potential strategy to kill cancer cells in which the repair of DSBs is deficient.

In this manuscript, Dagg et al. found by an in-depth analysis of the cancer genome atlas (TCGA) that there is a mutual exclusive signature between oncogene amplification (overexpression) and BRCA1/2 mutations. Searching for the underlying mechanisms operating the pathways, the authors found that WNT hyperactivation sensitizes BRCA1/2-deficient cells to death. The authors discovered and concluded that a beta-catenin overexpression suppressed p21 and thus promoted unscheduled replication firing and cell proliferation. The clash of stalled replication forks with a deficiency of DSB repair caused by BRCA1/2 induced cell death. They propose that targeting WNT or a specific oncogene may present a new approach as a pharmaceutical strategy for treating BRCA-deficient cancers. Overall, their findings are novel and useful for the field of replication and cell cycle control, and also potentially for cancer treatment. However, several points should be addressed, which I believe can make a strong case for publication in Nat Comms.

Major comments

1. The major mediator of WNT-mediated cell killing in BRCA1/2 deficient cells is p21. The authors found that beta-catenin stability downregulated the expression of the p21 gene (CDKN1A), while upregulated other genes (e.g., CycE, CDC6). The P21 expression is always high in BRCA1/2-deficient cancers. A reverse correlation of beta-catenin and the expression of the p21 gene (CDKN1A) is very interesting and needs more mechanistic understanding. The authors should demonstrate the mechanism by which b-catenin specifically upregulates CDKN1A. For example, one should examine the exon expression profile of CDKN1A and also check beta-catenin binding properties around, or breaks within, the CDKN1A locus and in correlation with intragenic firing.
2. The authors used b-catenin as a reference of WNT activity (with good argument) and the data are convincing. However, to make the finding as a general rule for "oncogene-mediated replication stress" it needs additional oncogenes tested. Otherwise, the authors should tune down and precisely describe what they have discovered about WNT oncogene-mediated lethality in BRCA1/2-deficient cancer cells. These points should be carefully formulated or rephrased for the title and abstract (line 50) and also in the main text.
3. The main claim for the "synthetic lethality" is based mainly on Fig 1c-e. The killing effect is obvious. However, the killing is more like a "synergistic" rather than a "synthetic" effect. Therefore, the description should be corrected. In addition, these curves are linear scale, but log scale should be presented.
4. The doxycyclin (DOX)-mediated inducible deletion of BRCA1/2 is used. The mark of the figs needs to be accurate: All figures where BRCA1/2 "+" or "-" should be Dox "-" or "+" respectively.
5. Fig 2d. The p21 pattern looks to be independent from the BRCA2 status. Fig 7c shows high p21 in the absence of BRCA2. Both blots need quantification in order to support the claim in respective texts.
6. Fig 3b. CycE does not seem to be upregulated by LY, especially not in BRCA2-deficient cells, while CDC6 seems to be higher. The statement is not correct (see line 250). The quantification is needed.
7. Fig 6d. Is there a difference between the BRCA2 "+" and the "-" samples? If there is a statistical difference, then BRCA2 has an impact on the survival in response to LY and RNase H1.
8. Fig 1c-d. 3a, ExData Fig 3a, 6a, 9. The P values should be provided.
9. It is generally not marked that how many experiments were repeated. This information is required for all figs.

10. Typos, e.g., lines 199, 213.

Reviewer #2 (Remarks to the Author):

In the manuscript entitled "A transcription-based mechanism for oncogene-induced lethality in BRCA1/2-deficient cells", Dagg and co-workers show that activation of WNT signaling is synthetically lethal with BRCA1/2 deficiency. The authors claim that an important underlying mechanism is the inhibition of CDKN1A transcription, and subsequent acceleration of G1/S transition.

This manuscript is based on the observation that BRCA1/2 mutations are mutually exclusive with a panel of oncogenes. The premiss here would be that oncogene activation leads to a specific type of DNA lesion, that requires BRCA1/2 for its resolution.

Mechanistically, the authors continue on the recent work by Thanos Halazonetis, which showed that oncogene activation leads to de novo replication initiation within gene bodies, and subsequent transcription-replication collisions.

In this manuscript, that model is projected onto cells with a BRCA2 defect. Taken together, the authors show that wnt activation through GSK3 inhibition leads to decreased survival (which was published previously by the authors). Beyond the previous finding, it is shown that GSK3 inhibition promotes G1/S transition, which goes along with altered origin usage. Finally, the authors propose p21 downregulation as an underlying mechanism that increases fork collapse and underlies toxicity.

The manuscript has some interesting data, but also has some conceptual and technical flaws. Firstly, the oncogene analysis is done on a very biased subset, whereas conclusions are made more generically.

Secondly, the models that are used are substandard to justify conclusions about the mutual exclusive nature of BRCA mutations and oncogenic WNT signaling. The approach to experimentally mimic oncogenic b-catenin is based on a chemical inhibitor that is ultimately toxic in all cell lines tested, which does not reflect the physiological proliferation and survival stimulating roles that oncogenic b-catenin is known for. It would be good to use models that reflect proper BRCA1/2 models, with a more relevant tissue context, and use stable genetic means to activate WNT signaling. In addition, many controls are missing, and data should be more rigorously analysed to support their conclusions.

Specific comments:

- The mutual exclusive nature of BRCA1/2 mutations and oncogene activation is only addressed for a limited panel of oncogenes, but is generalized in the conclusions (eg line 441/442). An important oncogene that is not included in this panel is MYC (but also other relevant breast/ovarian oncogenes should be included). As part of the previous work by Halazonetis, activation of both CCNE1 and MYC leads to excessive firing of replication sites within genes. However, MYC is not included in the oncogene panel in Suppl. Fig 1, likely because it significantly co-occurs with BRCA1 mutations. The authors should analyse a more comprehensive oncogene panel, and experimentally address if their model is limited to WNT activation. I also don't understand the statistics in extended figure 1. CCNE1 amplifications are 100% mutual exclusive, but only show limited statistical significance? Also for CCND1 amplification in ovarian cancer, there does not seem to be a corresponding elevation of CCND1 mRNA. It would be probably better to re-analyze tumor samples to exclude gains vs amps?

- The LY2090314 inhibitor should mimic oncogenic B-catenin. However, even in an established cancer model (H1299) treatment with LY2090314 is toxic (Fig 1). Similar for the RPE1 cell line models: there is a difference in sensitivity, but ultimately all cells are sensitive to treatment with LY2090314. This does not represent an oncogenic signal, nor does it reflect clean synthetic lethality. It would be better to test the hypothesis using genetically stable systems. Also, all data is plotted as relative survival, which makes it impossible to judge absolute differences between BRCA2 proficient and deficient.

- Similar to the point above: largest effects are observed upon acute knockdown of BRCA2. There are not many established human cancer cell line models with stable BRCA2 inactivation. One of them is the DLD1 cell line, which has been described as an isogenic BRCA2^{+/+} versus BRCA2^{-/-} cell lines, and is used by the authors in the extended figure. Notably, DLD1 harbours an APC mutation, leading to constitutive active b-catenin signaling, basically discarding the proposed model of synthetic lethality.

- The upregulation of CDC6, cyclin E upon LY2090314 treatment seems specific to the doxycycline-treated cells (Fig 2d), whereas LY2090314 treatment also leads to b-catenin accumulation in the -dox cells, and leads to similar gene expression changes in the -dox cells (extended data 4). Unclear what explains these differences, and why the effects on cdc6 and cyclin E are only observed in BRCA2-depleted cells.

- Figure 4: in contrast to previous work by the Halazonetis lab, origin usage does not seem to be at de novo loci, but at previously used sites, but sometimes at a higher frequency (ie: the red peaks are at sites of blue peaks). The text suggests that the replication events are 'b-catenin-induced', which to my opinion overinterprets the data.

There also appear to be origins that are more frequently used in the control situation (blue peaks with lower red peaks in Casz1 and ephB2 gene examples). These events should also be incorporated in the analysis.

- To appreciate the effect of GSK3 inhibition in Figure 5a, the DMSO condition should be included as a control, as well as the BRCA2-proficient cells.

- Figure 5C: usage of alternative exons: these changes seem very minor, and should be put in context of other genes, control settings (brca2-proficient), and statistics should be applied.

- The role of p21 is not very much substantiated. Can the authors show p21 levels in the cell experiments in extended figure 2? What is the interplay between acute BRCA loss and the ability of GSK3 inhibition to downregulate p21?

- Data not shown (line 355/366): please include data in extended data

- In line 220, it is stated that CDKN1A is downregulated by b-catenin. This is not evident from the RNAseq data. To me it is unclear how the authors came to the conclusion of investigating CDKN1A based on these data.

- The quality of the western blots of p21 is not impressive. Only effect of p21 on survival is tested in Figure 7, without showing images, and only showing one p21 siRNA. This is substandard. A second p21 siRNA should be used, and preferably a stable knockdown using shRNA, in which p21 is downregulated for the duration of the experiment. Similar experiments should be done using the cell lines models in extended figure to validate these results.

- The RNaseH1 experiment is done in HeLa cells, which makes it difficult to reconcile the R-loop data with the p21 data in the H1299 cells.

- In figure 5a: important controls are lacking in this experiment-

- The authors frequently use 'b-catenin activation' whereas 'GSK3 inhibition', or 'treatment with LY2090314' is a more accurate description. (eg lines 190, 208, 220, etc)

- Concerning data representation: some data are described to be representative of 3 independent experiments, but not standard deviations are shown (1c, 3a), or there appears to be no variation between experiments, which is unexpected (figure 1e, red dots).

-

- Figure 5b: mRNA expression data for H1299 would be better as a source of all expressed genes.

- There is a vertical line in blot BRC2 in figure 1b. is the blot cropped? If yes, is this also done for the other blots?

- Line 250 Cyclin E is with capital C.

- Line 250 This is confusing: "more significantly upregulated" I think the authors mean to say was 'upregulated', since a statistical test is lacking. Furthermore, I do not know what the authors are comparing this to: more upregulated compared to what? this is not clear from the text

- Regarding Figure 7A. The authors suggest that increase in replication fork speed in BRCA2-deficient cells results in more DNA lesions. The authors have a system in which fork velocity could be rescued using siRNA knock down of B-catenin. Could the authors show that increased fork velocity +/- GSK3i indeed results in more DNA lesions in their system (currently only shown for siRNA for p21) and could this be rescued by rescuing fork velocity.

- Line 431/433 results over overinterpreted here. "Stalled forks can be re-started and the DNA damage repaired using BRCA1/2-dependent mechanisms." The authors do not show that the stalled forks are restarted and that this is not the case in BRCA1/2 deficient cells. Line 433 " However, cells with compromised BRCA1/2 function lack effective means for fork restoration and DNA repair, and consequently accumulate collapsed forks and DNA damage, which trigger cell death." The authors claim that accumulation of B-catenin increase the amount of collapsed forks in BRCA deficient cells however the experiments that shows more collapsed forks is missing in the manuscript.

Reviewer #3 (Remarks to the Author):

In this manuscript, Dagg and colleagues show that mutations in BRCA1/2 genes and oncogene activation, two important drivers of tumorigenesis, are mutually exclusive in breast and ovarian cancers. This observation is based on the analysis of publicly available cancer genome data (TCGA and MSK-IMPACT) showing that BRCA1 and BRCA2 mutations are mutually exclusive with mutations in CCND1, CCNE1, CDC6 and BRAF (TCGA data) and in genes of the WNT/ β -catenin pathway (MSK-IMPACT). Next, the authors showed that the upregulation of the WNT/ β -catenin pathway through the chemical inhibition of the GSK3 β kinase was toxic to BRCA1- and BRCA2-deficient cell lines. To further characterize this novel synthetic lethal interaction between β -catenin activation and BRCA1/2 deficiency, Dagg et al. explored changes in gene expression in these cells. They found that the expression of the CDK inhibitor p21 is reduced and that several E2F target genes were upregulated, leading to an accelerated entry into S phase. This deregulated S phase was associated with the activation of new replication origins and with increased replication speed, leading to fork collapse and DNA damage. Of note, a large fraction of the new origins in cells overexpressing β -catenin were located within gene bodies and induced conflicts with transcription, which is consistent with earlier work from the Halazonetis lab showing that oncogene-induced origins induce replication stress. Based on these findings, the authors propose a model in which low p21 levels in cells overexpressing β -catenin cause a premature entry into S phase and a deregulation of both origin usage and fork speed, leading to increased replication stress. In the absence of BRCA1/2, cells would not be able to cope with the increased rate of fork stalling and collapse, which would explain why BRCA1/2-deficient cells do not tolerate oncogene-induced replication stress and rely on high p21 levels for growth. Overall, the data presented in this manuscript are of high quality and support the proposed model. This finding is novel, of broad interest and significance. However, important issues need to be addressed.

1. The authors convincingly show that the premature entry into S phase mediated by the upregulation of β -catenin leads to the activation of novel origins at a subset of genes. However, this effect is stronger in BRCA-deficient cell lines than in BRCA-proficient cells (Extended data Fig. 7a,b). Why is it so? In the examples shown figure 4, the β -catenin-induced origins seem to be active in control cells, albeit to a much lower level, which differs from CycE overexpressing cells (Macheret, 2018). Please comment on this difference. Moreover, it is not clear from the main text and the figure legend what 'sigma' corresponds to and why part of the data is masked in the

scatter plot (Fig. 4d). Finally, the intensity of the EdU signal at β -catenin-induced origins is much weaker compared to constitutive origins (Fig. 4e). This difference is not observed in the four examples shown above. Please explain.

2. BRCA2-deficient cells treated with the GSK3 β inhibitor LY2090314 show altered exon usage at a subset of genes containing β -catenin-induced origins (Fig. 5). Although this observation is consistent with the author's interpretation that aberrant origin firing interferes with gene expression, Dagg et al. do not provide direct evidence that altered replication impacts on exon usage in these cells. Their conclusion should therefore be toned down as the evidence is only correlative at this stage.

3. The experiment showing that LY2090314-treated BRCA2-deficient cells grow better when RNase H1 is overexpressed strongly supports the author's view that R-loops contribute to the synthetic lethal interaction between β -catenin activation and BRCA1/2 deficiency. However, the differences in S9.6 signals observed by immunofluorescence are fairly modest (Fig.6b) and the signal is only partially sensitive to RNase H1 in BRCA2-deficient cells (Fig. 6c). The authors would make their case stronger by using a complementary approach such as slot blotting or DRIP. Moreover, they need to indicate in the figure legend that nucleoli were excluded from analyses to eliminate non-specific signals.

4. The Carrera lab has shown more than a decade ago that PIK3 β regulates forks speed in mouse cells in a PKB-dependent manner by regulating p21 binding to PCNA (PMID: 19416922). Since PIK3 β and PKB also repress GSK3 β and upregulate β -catenin activity, this study is fully consistent with the results presented by Dagg and colleagues and should be discussed here.

Minor points:

Fig.1: There is a huge difference between the LY2090314 concentrations used in c and d panels. Does it reflect differences between cell lines or p53 status?

Line 96: The resolution of EdU-seq is fairly limited compared to other assays such as SNS-seq or OK-seq. 'high-resolution' should therefore be replaced with the resolution of the assay in kb.

Line 199: Remove 'we performed'

Line 274: RPA32 phosphorylation at Ser33

Lines 355-358: The experiments referred to as 'data not shown' should either be displayed or omitted.

Line 383: The recent work of the Penengo lab showing that accelerated forks induce DNA damage (PMID: 32597933) should also be cited.

Line 440: Is 'for the first time' really necessary?

We, the authors, are grateful to the Referees for their constructive comments on our manuscript. We have addressed all the points raised by the Referees, which significantly improved our manuscript.

Point-by-point response:

Reviewer #1:

Mutations in BRCA1/2 genes are associated with various human cancers affecting breast, ovarian, prostate and other tissues. BRCA1/2 handles the replication stress by stabilising stalled replication forks and conducting homologous recombination repair (HRR) permitting cell cycle progression. A deficiency of BRCA1/2 causes a collapse of the stalled replication forks thereby promoting cytotoxic double strand breaks (DSBs), which ultimately lead to genomic instability and the development of cancer. Oncogene overexpression accelerates the proliferation of (cancer) cells, namely elevated replication stress. Therefore, the coordination of the progression of replication in combination with a proficient repair of the stalled replication forks, is key to prevent genomic instability, cell cycle block and cell death. In case of BRCA1/2 deficiency, oncogene overexpression is even more cytotoxic to cells, which is likely due to the clash of unprogrammed replication firing and a deficiency of DSB repair. This approach, thus, may be considered as a potential strategy to kill cancer cells in which the repair of DSBs is deficient.

In this manuscript, Dagg et al. found by an in-depth analysis of the cancer genome atlas (TCGA) that there is a mutual exclusive signature between oncogene amplification (overexpression) and BRCA1/2 mutations. Searching for the underlying mechanisms operating the pathways, the authors found that WNT hyperactivation sensitizes BRCA1/2-deficient cells to death. The authors discovered and concluded that a beta-catenin overexpression suppressed p21 and thus promoted unscheduled replication firing and cell proliferation. The clash of stalled replication forks with a deficiency of DSB repair caused by BRCA1/2 induced cell death. They propose that targeting WNT or a specific oncogene may present a new approach as a pharmaceutical strategy for treating BRCA-deficient cancers. Overall, their findings are novel and useful for the field of replication and cell cycle control, and also potentially for cancer treatment. However, several points should be addressed, which I believe can make a strong case for publication in Nat Comms.

Major comments:

1. *The major mediator of WNT-mediated cell killing in BRCA1/2 deficient cells is p21. The authors found that beta-catenin stability downregulated the expression of the p21 gene (CDKN1A), while upregulated other genes (e.g., CycE, CDC6). The P21 expression is always high in BRCA1/2-deficient cancers. A reverse correlation of beta-catenin and the expression of the p21 gene (CDKN1A) is very interesting and needs more mechanistic understanding. The authors should demonstrate the mechanism by which b-catenin specifically upregulates CDKN1A. For example, one should examine the exon expression profile of CDKN1A and also check beta-catenin binding properties around, or breaks within, the CDKN1A locus and in correlation with intragenic firing.*

Response: We would like to thank the Reviewer for this suggestion, which is indeed excellent. We searched publicly available ChIP-seq datasets and found no evidence that β -

catenin or its interacting transcription factors (LEF-1, TCF) can bind directly to the *CDKN1A* promoter. We therefore examined the possibility that *CDKN1A* expression is regulated by β -catenin indirectly, via an effector, and we used a candidate approach. MYC is known to inhibit transcription of *CDKN1A* gene by directly binding to its promoter (Coller *et al.*, PNAS 2000; Gartel *et al.*, PNAS 2001; reviewed in Garcia-Gutierrez *et al.*, Genes 2019) and MYC protein is stabilised by β -catenin accumulation (Xu *et al.*, Mol Carcinogenesis 2016). Therefore, we tested the hypothesis that β -catenin accumulation leads to increased MYC protein levels, and this, in turn, inhibits the expression of p21 in our cellular model for inducible BRCA2 inactivation (Supplementary Fig. 10b). Our results, included in Supplementary Fig. 10c, indicate that β -catenin depletion reduced the expression levels of MYC protein, consistent with previously published work. We further demonstrate that siRNA-mediated MYC depletion, as well as MYC chemical inhibition, increased p21 protein levels in both BRCA2-proficient and -deficient cells (Supplementary Fig. 10d,e), supporting an indirect p21 suppression by β -catenin, mediated by increased MYC protein levels. Description of these results is included on p. 13-14 of the revised manuscript.

2. The authors used b-catenin as a reference of WNT activity (with good argument) and the data are convincing. However, to make the finding as a general rule for “oncogene-mediated replication stress” it needs additional oncogenes tested. Otherwise, the authors should tune down and precisely describe what they have discovered about WNT oncogene-mediated lethality in BRCA1/2-deficient cancer cells. These points should be carefully formulated or rephrased for the title and abstract (line 50) and also in the main text.

Response: We agree with the Reviewer that the data presented in the previous version of our manuscript refer to β -catenin alone. In response to the Reviewer's suggestion, we extended the oncogene-mediated toxicity in the context of BRCA1/2 inactivation to overexpression of two other oncogenes: cyclin D1 and cyclin E. To address this, we generated a U2OS cell line with inducible (Tet-ON) overexpression of cyclin D1 (Supplementary Fig. 4a). Cyclin D1 overexpression accelerated S-phase entry (Supplementary Fig. 4b) and is therefore oncogenic, as previously reported for the cyclin E oncogene (Macheret & Halazonetis, Nature 2018) and as shown for β -catenin in Fig. 3a and Supplementary Fig. 7a of our paper. Both cyclin D1 and cyclin E overexpression substantially reduced proliferation of cells lacking BRCA1 or BRCA2 (Supplementary Fig. 4c,d). These results are discussed on p. 7 of our revised manuscript.

3. The main claim for the “synthetic lethality” is based mainly on Fig 1c-e. The killing effect is obvious. However, the killing is more like a “synergistic” rather than a “synthetic” effect. Therefore, the description should be corrected. In addition, these curves are linear scale, but log scale should be presented.

Response: We agree with the Reviewer and have replaced “synthetic lethality” with “increased toxicity/synergy” in the revised version of our manuscript. In Fig. 1c, d, e we used \log_{10} scales for the x axis.

4. The doxycyclin (DOX)-mediated inducible deletion of BRCA is used. The mark of the figs needs to be accurate: All figures where BRCA “+” or “-” should be Dox “-“ or “+” respectively.

Response: We believe that our data labelling style (+/-BRCA2, rather than -/+DOX) makes it easier for the reader to follow our manuscript. We therefore kept the existing labels and we included in all figure legends the explanation that +/-BRCA2 labels indicate -/+DOX.

5. Fig 2d. The p21 pattern looks to be independent from the BRCA2 status. Fig 7c shows high p21 in the absence of BRCA2. Both blots need quantification in order to support the claim in respective texts.

Response: In response to the Reviewer's suggestion, we included the quantification of p21 signal relative to GAPDH loading control in Fig. 2d and Fig. 7c showing p21 immunoblots. This quantification demonstrates that p21 expression is higher in BRCA2-deficient cells relative to BRCA2-proficient control cells.

6. Fig 3b. CycE does not seem to be upregulated by LY, especially not in BRCA2-deficient cells, while CDC6 seems to be higher. The statement is not correct (see line 250). The quantification is needed.

Response: In response to the Reviewer's request, we included quantification of cyclin E and CDC6 signal in Fig. 3b. In addition, we clarified in the text that β -catenin (LY)-dependent cyclin E induction was only detected in the early timepoints after release from mitotic arrest (6 and 8 hours), whilst increased CDC6 expression was detected at all timepoints examined (p. 9 of the revised manuscript).

7. Fig 6d. Is there a difference between the BRCA2 "+" and the "-" samples? If there is a statistical difference, then BRCA2 has an impact on the survival in response to LY and RNase H1.

Response: As the Reviewer indicated, there is a statistical difference between BRCA2 "+" and "-" samples (**, $P < 0.01$) which is now indicated in the new Fig. 6d. We have explained in the figure legend that these values were expressed relative to DMSO-treated +BRCA2 or -BRCA2 samples, which were set to 100%. We changed y axis labelling to indicate this.

8. Fig 1c-d. 3a, ExData Fig 3a, 6a, 9. The P values should be provided.

Response: In the revised version of our manuscript, we included P values for all the figures indicated. For the data shown in Supplementary Fig. 10 a (previously Supplementary Fig. 9) the lack of statistical significance is likely due to the low number of tumours available in TCGA for these analyses (indicated in the revised manuscript on p. 13). Although the number of samples was low, we were nevertheless able to detect an inverse correlation between the mean expression of *CTNNB1* gene encoding β -catenin and the mean expression of *CDKN1A* gene encoding p21.

9. It is generally not marked that how many experiments were repeated. This information is required for all figs.

Response: The number of independent experiments is now indicated in each figure legend. For EdUseq, we performed at least two independent experiments for each genotype under similar conditions (now indicated in Experimental Procedures).

10. Typos, e.g., lines 199, 213.

Response: We thank the Reviewer for pointing out these typos, which have been corrected in the revised version of our manuscript.

Reviewer #2:

In the manuscript entitled “A transcription-based mechanism for oncogene-induced lethality in BRCA1/2-deficient cells”, Dagg and co-workers show that activation of WNT signaling is synthetically lethal with BRCA1/2 deficiency. The authors claim that an important underlying mechanism is the inhibition of CDKN1A transcription, and subsequent acceleration of G1/S transition.

This manuscript is based on the observation that BRCA1/2 mutations are mutually exclusive with a panel of oncogenes. The premiss here would be that oncogene activation leads to a specific type of DNA lesion, that requires BRCA1/2 for its resolution. Mechanistically, the authors continue on the recent work by Thanos Halazonetis, which showed that oncogene activation leads to de novo replication initiation within gene bodies, and subsequent transcription-replication collisions.

In this manuscript, that model is projected onto cells with a BRCA2 defect. Taken together, the authors show that wnt activation through GSK3 inhibition leads to decreased survival (which was published previously by the authors). Beyond the previous findings, it is shown that GSK3 inhibition promotes G1/S transition, which goes along with altered origin usage. Finally, the authors propose p21 downregulation as an underlying mechanism that increases fork collapse and underlies toxicity.

Response: We thank the Reviewer for taking the time to read and comment on our manuscript. However, it is incorrect to state that our results showing that GSK3 inhibition decreased survival of BRCA1/2-deficient cells have been published before. These results are reported for the first time in the current manuscript. As such, there are no ‘previous findings’ to relate the current manuscript to.

The manuscript has some interesting data, but also has some conceptual and technical flaws. Firstly, the oncogene analysis is done on a very biased subset, whereas conclusions are made more generically.

Response: We thank the Reviewer for pointing out the generalisation of our findings as reported in the previous version of our manuscript. As explained in point 2 raised by Reviewer 1, we have now included cyclin D1 and cyclin E as additional models for oncogene overexpression-induced toxicity in the context of BRCA1/2 abrogation (new Supplementary Fig. 4), to support the general value of our findings. These results are discussed on p. 7 of our revised manuscript.

Secondly, the models that are used are substandard to justify conclusions about the mutual exclusive nature of BRCA mutations and oncogenic WNT signaling. The approach to experimentally mimic oncogenic b-catenin is based on a chemical inhibitor that is ultimately toxic in all cell lines tested, which does not reflect the physiological proliferation and survival stimulating roles that oncogenic b-catenin is known for. It would be good to use models that reflect proper BRCA1/2 models, with a more relevant tissue context, and use stable genetic means to activate WNT signaling. In addition, many controls are missing, and data should be more rigorously analysed to support their conclusions.

Response: It is well-established that WNT signalling activation leads to β -catenin stabilisation via suppression of its GSK3-dependent phosphorylation. The GSK3 inhibitor used in our study inhibits GSK3 catalytic activity and the β -catenin phosphorylation at Ser33/Ser37/Thr41, thereby preventing its degradation. These results are clearly demonstrated in our paper (Fig. 1b). Moreover, we recapitulated GSK3 chemical inhibition using genetic tools (siRNAs against GSK3a/b; Supplementary Fig. S3), which also decreased proliferation of BRCA1/2-deficient cells. Finally, one of the cellular models for BRCA1/2 inactivation used in our study are MDA-MB-231 breast tumour-derived cells (e.g. Fig. 7d, Supplementary Fig. 2c,e), therefore our results are validated in relevant tissue context.

Specific comments:

1. The mutual exclusive nature of BRCA1/2 mutations and oncogene activation is only addressed for a limited panel of oncogenes, but is generalized in the conclusions (eg line 441/442).

Response: Please see above our response to point 2 raised by Reviewer 1.

An important oncogene that is not included in this panel is MYC (but also other relevant breast/ovarian oncogenes should be included). as part of the previous work by Halozonetis, activation of both CCNE1 and MYC leads to excessive firing of replication sites within genes. However, MYC is not included in the oncogene panel in Suppl. Fig 1, likely because it significantly co-occurs with BRCA1 mutations.

Response: In response to the Reviewer's comment, we included new results in Figure below showing that MYC amplification co-occurs with BRCA1/2 inactivation in breast and ovarian tumours (mutual exclusivity test shows lack of statistical significance). These results recapitulate those reported by Annunziato *et al.*, Nat. Comm. 2019 and are now discussed on p. 16-17 of our revised manuscript.

Figure for Reviewer: Lack of mutual exclusivity between BRCA1/2 mutations and MYC gene alterations in breast and ovarian tumours. a,b, Alterations of BRCA1, BRCA2 and MYC in the TCGA breast³⁶ (a) and TCGA ovarian tumour collections²⁷ (b). Only tumours with alterations in one of the indicated genes are

shown. %, frequency of the indicated alterations in each gene. Mutual exclusivity tests between alterations in *BRCA1* and *BRCA2* genes, and amplification of *MYC* oncogene were performed using the DISCOVER method¹⁰⁰. **c,d**, mRNA expression of *MYC* (z-scores) in breast (**c**) or ovarian (**d**) tumours with no alterations or with amplifications (AMP) in the respective oncogenes. ***, $P < 0.001$; ****, $P < 0.0001$ (Mann-Whitney test). **e**, Alterations of *MYC* in the MSK breast tumour collection³⁷. Only tumours with alterations in one of the indicated genes shown. %, frequency of the indicated alterations in each gene. Mutual exclusivity tests between alterations in *BRCA1*, *BRCA2* and *MYC* genes were performed as in (**a** and **b**). NS, $P > 0.05$.

I also don't understand the statistics in extended figure 1. CCNE1 amplifications are 100% mutual exclusive, but only show limited statistical significant? Also for CCND1 amplification in ovarian cancer, there does not seem to be a corresponding elevation of CCND1 mRNA. It would be probably better to re-analyze tumor samples to exclude gains vs amps?

Response: Regarding *CCNE1*, the number of tumours with *CCNE1* amplification in this dataset is very low, which explains the limited statistical significance in Supplementary Fig. 1. Regarding *CCND1*: its amplification is used as a biomarker for breast, but not ovarian cancers (Bell *et al.*, Nature 2011; Patch *et al.*, Nature 2015), although it shows mutual exclusivity with *BRCA1/2* gene inactivation in both types of tumours (Supplementary Fig.

S1). As the Reviewer suggested, we tested the impact of *CCND1* gains vs amplifications in ovarian tumours and found that the mRNA levels are significantly enhanced in tumours with *CCND1* amplification, but not in tumours with *CCND1* copy number gains (Figure below).

Figure for Reviewer: *CCND1* expression in ovarian tumours with *CCND1* copy gain and amplification. mRNA expression of *CCND1* (z-scores) in ovarian tumours

with no alterations ($n = 136$), copy number gains ($n = 117$) or with amplifications (AMP; $n = 13$). NS, $P > 0.05$; *, $P < 0.05$ (Mann-Whitney test).

2. *The LY2090314 inhibitor should mimic oncogenic B-catenin. However, even in an established cancer model (H1299) treatment with LY2090314 is toxic (Fig 1). Similar for the RPE1 cell line models: there is a difference in sensitivity, but ultimately all cells are sensitive to treatment with LY2090314. This does not represent an oncogenic signal, nor does it reflect clean synthetic lethality. It would be better to test the hypothesis using genetically stable systems. Also, all data is plotted as relative survival, which makes it impossible to judge absolute differences between BRCA2 proficient and deficient.*

Response: As indicated by the P values included in the new Fig. 1, LY2090314 toxicity is higher in the context of *BRCA1/2* deficiency than in wild-type cells. In the latter, LY2090314 is toxic at higher concentrations. In response to the Reviewer's request, also requested by Reviewer 1, point 3, we have replaced "synthetic lethality" with "increased toxicity/synergy" in the revised version of our manuscript. Plotting survival data relative to untreated cells of the same genotype is the most common representation of data from clonogenic survival assays.

3. *Similar to the point above: largest effects are observed upon acute knockdown of BRCA2.*

There are not many established human cancer cell line models with stable BRCA2 inactivation. One of them is the DLD1 cell line, which has been described as an isogenic BRCA2+/+ versus BRCA2-/- cell lines, and is used by the authors in the extended figure. Notably, DLD1 harbours an APC mutation, leading to constitute active b-catenin signaling, basically discarding the proposed model of synthetic lethality.

Response: We agree with the Reviewer that DLD1 cells have intrinsically high β -catenin levels. We speculate that these cells have conceivably adapted to high β -catenin, thus providing an explanation for why they tolerate BRCA2 deletion. In spite of their high endogenous β -catenin levels, our new data included in Supplementary Fig. S2d show that treatment with LY2090314 further increased β -catenin expression in these cells. Consequently, BRCA2^{-/-} DLD1 cells are also sensitive to LY2090314 treatment, thus supporting our model for β -catenin-induced toxicity in the absence of BRCA2. These results are discussed on p. 6 of our revised manuscript.

4. The upregulation of CDC6, cyclin E upon LY2090314 treatment seems specific to the doxycycline-treated cells (Fig 2d), whereas LY2090314 treatment also leads to b-catenin accumulation in the -dox cells, and leads to similar gene expression changes in the -dox cells (extended data 4). Unclear what explains these differences, and why the effects on cdc6 and cyclin E are only observed in BRCA2-depleted cells.

Response: LY2090314 treatment does not lead 'to similar gene expression changes' in BRCA2-deficient and -proficient cells, as a high number of genes are deregulated specifically in the BRCA2-deficient setting (Figure 2b). Why certain genes are deregulated by β -catenin in BRCA2-deficient, but not in BRCA2-proficient cells is an interesting question. Multiple explanations can be envisaged, including differences in cell cycle progression between the two conditions. Testing these possibilities is beyond the scope of the current paper, although they will be addressed in the context of a future lab project.

5. Figure 4: in contrast to previous work by the halazonetis lab, origin usage does not seem to be at de novo loci, but at previously used sites, but sometimes at a higher frequency (ie: the red peaks are at sites of blue peaks). The text suggests that the replication events are 'b-catenin-induced', which to my opinion overinterprets the data. there also appear to be origins that are more frequently used in the control situation (blue peaks with lower red peaks in Casz1 and ephB2 gene examples). These events should also be incorporated in the analysis.

Response: The events mentioned by the Reviewer are incorporated in analysis. Both in our paper, as in the paper by Macheret & Halazonetis (Nature 2018), oncogene-induced origins were defined as origins that fire at 1:2 to 1:4 ratio after oncogene activation relative to cells with "normal" oncogene expression. Indeed, there is always origin firing in the "normal" cells, which have lower levels of replication stress than the oncogene overexpressing cells.

6. To appreciate the effect of GSK3 inhibition in Figure 5a, the DMSO condition should be included as a control, as well as the BRCA2-proficient cells.

Response: DMSO-treated samples are included in the comparison described above, as 'normal' cells. BRCA2-proficient H1299 cells are included in the new Fig. 5b.

7. Figure 5C: usage of alternative exons: these changes seem very minor, and should be put in context of other genes, control settings (brca2-proficient), and statistics should be applied.

Response: EdUseq was performed in cells synchronised in mitosis and released in the cells cycle, whilst RNAseq was performed in asynchronous cells. The differences in experimental setup might indeed be reflected in the minor changes in exon usage indicated by the Reviewer. Since performing both experiments in identical conditions is not easily achievable, we decided to eliminate Fig. 5c, which does not compromise the conclusions of our paper.

8. *The role of p21 is not very much substantiated. Can the authors show p21 levels in the cell experiments in extended figure 2? What is the interplay between acute BRCA loss and the ability of GSK3 inhibition to downregulate p21?*

Response: In response to the Reviewer's suggestion, we included WB of p21 expression in BRCA2-proficient vs -deficient MDA-MB-231 cells treated with LY2090314 in Supplementary Fig. 2e.

9. *Data not shown (line 355/366): please include data in extended data*

Response: We included DRB treatment in Fig. 6a and EU quantification in BrdU-positive cells is shown in the new Supplementary Fig. 9a.

10. *In line 220, it is stated that CDKN1A is downregulated by b-catenin. This is not evident from the RNAseq data. To me it is unclear how the authors came to the conclusion of investigating CDKN1A based on these data.*

Response: Our RNA-seq data clearly indicate *CDKN1A* downregulation in BRCA2-deficient cells by β -catenin (Fig. 2a). Since p21 downregulation is crucial in promoting G1/S transition (reviewed by Dutta, 2005) and since the G1 arrest in BRCA2-deficient cells is overcome by β -catenin accumulation, p21 was an obvious candidate as a mediator of the β -catenin effect on the G1 arrest in these cells.

11. *the quality of the western blots of p21 is not impressive. Only effect of p21 on survival is tested in Figure 7, without showing images, and only showing one p21 siRNA. This is substandard. A second p21 siRNA should be used, and preferably a stable knockdown using shRNA, in which p21 is downregulated for the duration of the experiment. Similar experiments should be done using the cell lines models in extended figure to validate these results.*

Response: We have included results obtained with a second siRNA targeting p21 in Supplementary Fig. 11 (and p. 15 of the revised manuscript). These are consistent with the data reported in Fig. 7f,g.

12. *The RNaseH1 experiment is done in HeLa cells, which makes it difficult to reconcile the R-loop data with the p21 data in the H1299 cells.*

Response: The only cellular model for inducible RNaseH1 expression (generated by the laboratory of Karlene Cimprich, Stanford) is in HeLa cells. Such cell lines are difficult to generate due to the high toxicity of residual RNaseH1 expression. Our experiment in HeLa cells supports the concept that RNaseH1 expression abrogates S9.6 signal, which proves that S9.6 recognises DNA-RNA hybrids. The new data included in Supplementary Fig. 2f (and p. 12 of the revised manuscript) show that p21 suppression by LY2090314 treatment is dependent on β -catenin in HeLa cells, similarly to H1299 cells.

13. *In figure 5a: important controls are lacking in this experiment-*

Response: We addressed this in point 6 raised by the same Reviewer above.

14. The authors frequently use 'b-catenin activation' whereas 'GSK3 inhibition', or 'treatment with LY2090314' is a more accurate description. (eg lines 190, 208, 220, etc)

Response: The data included in our manuscript rigorously demonstrate that the effects on GSK3i on E2F targets (cyclin E and CDC6), as well as on S-phase entry and on *CDKN1A* transcription and expression are mediated by β -catenin. These experiments contained cells treated with β -catenin siRNA to confirm that the effects of GSK3i were β -catenin dependent.

15. Concerning data representation: some data are described to be representative of 3 independent experiments, but not standard deviations are shown (1c, 3a), or there appears to be no variation between experiments, which is unexpected (figure 1e, red dots).

Response: The SEM values shown in the figures indicated are very small, which is why it appears that there is no error bar.

16. Figure 5b: mRNA expression data for H1299 would be better as a source of all expressed genes.

Response: In the new Fig. 5c we include data using H1299 transcriptome (instead of the human transcriptome, hg19) which did not alter our conclusion that longer genes are more frequently affected by oncogene-induced origin firing.

17. There is a vertical line in blot BRC2 in figure 1b. is the blot cropped? If yes, is this also done for the other blots?

Response: Cropped blots are clearly indicated in the corresponding figures, including Figure 1b.

18. Line 250 Cyclin E is with capital C.

Response: This is corrected in the revised manuscript.

19. Line 250 This is confusing: "more significantly upregulated" I think the authors mean to say was 'upregulated', since a statistical test is lacking. Furthermore, I do not know what the authors are comparing this to: more upregulated compared to what? this is not clear from the text

Response: Upregulation was compared to cells lacking β -catenin induction. This was clarified in the text (p. 9), also being requested by Reviewer 1, point 6.

20. Regarding Figure 7A. The authors suggest that increase in replication fork speed in BRCA2-deficient cells results in more DNA lesions. The authors have a system in which fork velocity could be rescued using siRNA knock down of B-catenin. Could the authors show that increased fork velocity +/- GSK3i indeed results in more DNA lesions in their system (currently only shown for siRNA for p21) and could this be rescued by rescuing fork velocity.

Response: It was previously shown (Maya-Mendoza et al., Nature 2019) that increased fork velocity can induce DNA lesions, therefore we did not question their result. In Fig. 3e we show that GSK3i triggers DNA damage. We were unable to examine whether this could be abrogated by β -catenin depletion, as β -catenin siRNA treatment increased the frequency of 53BP1 foci.

21. Line 431/433 results over interpreted here. "Stalled forks can be re-started and the DNA damage repaired using BRCA1/2-dependent mechanisms." The authors do not show that the stalled forks are restarted and that this is not the case in BRCA1/2 deficient cells. Line 433 " However, cells with compromised BRCA1/2 function lack effective means for fork restoration and DNA repair, and consequently accumulate collapsed forks and DNA damage, which trigger cell death." The authors claim that accumulation of B-catenin increase the amount of collapsed forks in BRCA deficient cells however the experiments that shows more collapsed forks is missing in the manuscript.

Response: We agree with the Reviewer that collapsed forks accumulate in both BRCA2-proficient and -deficient cells as a result of LY2090314 treatment. However, we only detect RPA phosphorylation at Ser33 in cells lacking BRCA2 (Fig. 3d), suggesting that degradation of β -catenin-induced stalled/collapsed forks may be more pronounced when BRCA2 is abrogated. We have amended the interpretation of our results on lines 431/433 (currently on p. 11) to reflect this.

Reviewer #3:

In this manuscript, Dagg and colleagues show that mutations in BRCA1/2 genes and oncogene activation, two important drivers of tumorigenesis, are mutually exclusive in breast and ovarian cancers. This observation is based on the analysis of publicly available cancer genome data (TCGA and MSK-IMPACT) showing that BRCA1 and BRCA2 mutations are mutually exclusive with mutations in CCND1, CCNE1, CDC6 and BRAF (TCGA data) and in genes of the WNT/ β -catenin pathway (MSK-IMPACT). Next, the authors showed that the upregulation of the WNT/ β -catenin pathway through the chemical inhibition of the GSK3 β kinase was toxic to BRCA1- and BRCA2-deficient cell lines. To further characterize this novel synthetic lethal interaction between β -catenin activation and BRCA1/2 deficiency, Dagg et al. explored changes in gene expression in these cells. They found that the expression of the CDK inhibitor p21 is reduced and that several E2F target genes were upregulated, leading to an accelerated entry into S phase. This deregulated S phase was associated with the activation of new replication origins and with increased replication speed, leading to fork collapse and DNA damage. Of note, a large fraction of the new origins in cells overexpressing β -catenin were located within gene bodies and induced conflicts with transcription, which is consistent with earlier work from the Halazonetis lab showing that oncogene-induced origins induce replication stress. Based on these findings, the authors propose a model in which low p21 levels in cells overexpressing β -catenin cause a premature entry into S phase and a deregulation of both origin usage and fork speed, leading to increased replication stress. In the absence of BRCA1/2, cells would not be able to cope with the increased rate of fork stalling and collapse, which would explain why BRCA1/2-deficient cells do not tolerate oncogene-induced replication stress and rely on high p21 levels for growth. Overall, the data presented in this manuscript are of high quality and support the proposed model. This finding is novel, of broad interest and significance. However, important issues need to be addressed.

1. The authors convincingly show that the premature entry into S phase mediated by the upregulation of β -catenin leads to the activation of novel origins at a subset of genes.

However, this effect is stronger in BRCA-deficient cell lines than in BRCA-proficient cells (Extended data Fig. 7a,b). Why is it so?

Response: The Reviewer is correct in pointing out that the effect of β -catenin upregulation on S-phase entry and activation of novel origins is stronger in BRCA2-deficient cells. We attributed this to the fact that cells lacking BRCA2 frequently arrest in G1 and show slower replication fork progression. This is stated in p. 10-11 of the revised manuscript.

In the examples shown figure 4, the β -catenin-induced origins seem to be active in control cells, albeit to a much lower level, which differs from CycE overexpressing cells (Macheret, 2018). Please comment on this difference.

Response: The Reviewer correctly observed that the β -catenin-induced origins also fire in DMSO-treated cells, although to much lower levels, and these levels are different in Cyclin E overexpressing cells. This inconsistency can be explained by the fact that these are different cell lines and different oncogenes. However, the definition of sigma value used to identify oncogene induced origins (twice to four times higher relative to “normal” cells) is the same in our study as in Macheret & Halazonetis, Nature 2018. We explained this also in point 5 raised by Reviewer 2.

Moreover, it is not clear from the main text and the figure legend what ‘sigma’ corresponds to and why part of the data is masked in the scatter plot (Fig. 4d). Finally, the intensity of the EdU signal at β -catenin-induced origins is much weaker compared to constitutive origins (Fig. 4e). This difference is not observed in the four examples shown above. Please explain.

Response: Regarding data in Fig. 4d: We require a threshold level of signal before we consider a site to be an origin. Therefore, the data are not masked. All origins defined as above are shown. Regarding data in Fig. 4e: on average, the oncogene-induced origins show a lower signal than the constitutive origins, because they do not fire in all cells, but only in the cells that enter S-phase quite fast (before the gene has been transcribed).

2. BRCA2-deficient cells treated with the GSK3 β inhibitor LY2090314 show altered exon usage at a subset of genes containing β -catenin-induced origins (Fig. 5). Although this observation is consistent with the author’s interpretation that aberrant origin firing interferes with gene expression, Dagg et al. do not provide direct evidence that altered replication impacts on exon usage in these cells. Their conclusion should therefore be toned down as the evidence is only correlative at this stage.

Response: We agree with the Reviewer’s point. As explained in response to Reviewer 2, point 7, the minor changes observed in exon usage could be due to the different experimental setup of our EdUseq and RNAseq analyses. Since performing both experiments in identical conditions is not trivial, we decided to eliminate previous Fig. 5c, without compromising the conclusions of our paper.

3. The experiment showing that LY2090314-treated BRCA2-deficient cells grow better when RNase H1 is overexpressed strongly supports the author’s view that R-loops contribute to the synthetic lethal interaction between β -catenin activation and BRCA1/2 deficiency. However, the differences in S9.6 signals observed by immunofluorescence are fairly modest (Fig.6b) and the signal is only partially sensitive to RNase H1 in BRCA2-deficient cells (Fig. 6c). The authors would make their case stronger by using a complementary approach such

as slot blotting or DRIP. Moreover, they need to indicate in the figure legend that nucleoli were excluded from analyses to eliminate non-specific signals.

Response: In response to the Reviewer's suggestion, we performed DRIP experiments, which are now included in the new Supplementary Fig. 9b,c,d. Our DRIP results indicate that the S9.6 signal at the specific loci analysed and previously validated (*BTBD19*, *APOE* and *RPL13A*) is sensitive to RNaseH1 and is enhanced by GSK3i in β -catenin-dependent manner. Furthermore, we indicated in the legend of Figure 6 that nucleoli were excluded from the analysis, as the Reviewer requested.

4. The Carrera lab has shown more than a decade ago that PIK3bregulates forks speed in mouse cells in a PKB-dependent manner by regulating p21 binding to PCNA (PMID: 19416922). Since PIK3 β and PKB also repress GSK3 β and upregulate β -catenin activity, this study is fully consistent with the results presented by Dagg and colleagues and should be discussed here.

Response: We thank the Reviewer for suggesting these references, which are indeed relevant and have been added to p. 14 and discussed in p. 18 of the revised manuscript.

Minor points:

Fig.1: There is a huge difference between the LY2090314 concentrations used in c and d panels. Does it reflect differences between cell lines or p53 status?

Response: The difference in the LY2090314 concentrations used in Fig. 1c,d reflect cell line-specific requirements, possibly dependent on the endogenous levels of β -catenin (before induction by LY2090314). We do not believe that p53 status plays a part, as both cell lines are p53-negative.

Line 96: The resolution of EdU-seq is fairly limited compared to other assays such as SNS-seq or OK-seq. 'high-resolution' should therefore be replaced with the resolution of the assay in kb.

Response: We agree with this point and have replaced "high-resolution" to "10 kb resolution" in p. 4 of the revised manuscript.

Line 199: Remove 'we performed'

Response: This was removed in the revised manuscript.

Line 274: RPA32 phosphorylation at Ser33

Response: We corrected this in the revised manuscript.

Lines 355-358: The experiments referred to as 'data not shown' should either be displayed or omitted.

Response: We have included the EU quantification in BrdU-positive cells in the new Supplementary Fig. 9a, as also requested by Reviewer 2, point 9.

Line 383: The recent work of the Penengo lab showing that accelerated forks induce DNA damage (PMID: 32597933) should also be cited.

Response: This reference was added in the revised manuscript.

Line 440: Is 'for the first time' really necessary?

Response: We agree with the Reviewer and we eliminated these words in the revised manuscript.

REVIEWERS' COMMENTS:

Reviewer #1 (Remarks to the Author):

The authors have taken significant steps to address my original comments. The revised work and manuscript are good work. The responses to reviewers' comments are reasonable and acceptable. I do not have further comments.

Reviewer #2 (Remarks to the Author):

Overall, the manuscript addresses an important topic and uses a highly innovative methodology to investigate the relation of oncogene expression and defective HR.

A comment that I originally raised, and which is only partially addressed, is that the conclusions and title are too generic, and should be toned down and be phrased more specifically to match the results of the experiments.

The far majority of the results reflect b-catenin upregulation, as also reflected in the final model that the authors present. I would highly recommend adjusting the title, abstract and start of the discussion accordingly, and mention that parts of these effects can be extended to other oncogenes.

Comments:

-The title and abstract state that oncogene amplification is mutually exclusive with BRCA1/2 inactivation. This is true only for the selected panel of oncogenes. The authors in their rebuttal letter indicate that MYC is not mutually exclusive with BRCA1/2 mutation. Of note, previous work by the authors showed that MYC and Cyclin E both trigger aberrant origin firing (Macharet, Nature, 2018). So, the conclusion cannot be that this is a generic mechanism explaining SL interactions between oncogene activation and BRCA defects.

Although the authors acknowledge this in the rebuttal letter, these data are not discussed in the manuscript or toned down the conclusions.

Also, the first part of the discussion still describes the results as a universal mechanism. To present a picture that does justice to the complexity of HR-defective tumors, the above-mentioned data should be part of the figures and discussion in the manuscript, not only the rebuttal letter.

I realize that text is added to the discussion on MYC. However, this part only provides a very limited interpretation of the available literature. In the updated discussion, Myc is described as an exceptional case, which works through triggering copy number variations, including MCL1. However, similar long-term effects driving CNA acquisition have been described for CCNE1 (eg. Negrini, 2010 and Aziz, 2019), which is not described to discard the model. More importantly, Myc has been shown to triggers de novo origin firing like CCNE1, in short-term experiments. Moreover, Myc is not only regulated at the transcriptional level by b-catenin, but is also amplified frequently, and is not mutual exclusive with BRCA1/2 mutations in tumor databases.

-concerning the lethal effects of oncogene expression in BRCA1/2 defective cells, Cyclin D and cyclin E are now included (Suppl. Figure 4), overexpression of which clearly induces S-phase entry, both in control and BRCA2-depleted cells.

The experiment in Suppl. 4c/d is difficult to interpret when the effects are only plotted as a separate relative measure in subcategories. BRCA1/2 inactivation is lethal in many cell lines, and these effects cannot be judged in these plots. These plots should ideally be visualized as absolute proliferation measures. Alternatively, the plots could be plotted as relative measures to only one condition (-dox/siControl) to appreciate all relevant effects in these experiments.

Also, the interpretation of the experiment should be toned down. Lower proliferation is not the same as viability or toxicity (Line 187/188: 'specifically toxic'; Line 190: 'decrease in viability', Line 479: 'the same toxicity';) but should be referred to as 'decreased proliferation'.

-a minor, but important issue: the labeling of the figures contains an interpretation of the experiment, rather than the actual procedure. '-BRCA2' or '+BRCA2' could indicate reconstitution, knockout etc, but in this situation reflects a dox-inducible shRNA. '+/- dox' is much closer to the

actual experiments (like in Fig1e, where siControl and siB-catenin is used properly, as well as in figures of Suppl. 4, where +/- dox is used). The same is true for 1C, where 'BRCA2-proficient' vs 'BRCA2-deficient' is used.

-line 134: 'oncogenic B-catenin activation' refers to treatment with a GSK3 inhibitor. It remains unclear if this reflects 'oncogenic' B-catenin activation. The correct way to state this would be: 'GSK3 inhibition is lethal with BRCA1 or BRCA2 inactivation' or 'b-catenin activation is lethal with BRCA1 or BRCA2 inactivation'. Similarly: 'its oncogenic expression' should be 'its expression'. Also applies to 'oncogenic' in titles of suppl. Fig 4 and 5.

-suppl. Figure 2c/e: BRCA2 blots are lacking, which are needed to interpret the experiment.

-line 1346: SEM cannot be derived from a single experiment. I guess this should be 'SEM of the three independent experiments'.

Reviewer #3 (Remarks to the Author):

The authors have significantly improved their manuscript and have addressed all the issues that I raised. In my opinion, this work is now suitable for publication in Nature Communications. Since the question of why intragenic oncogene-induced origins are active in control cells (albeit at a lower level) has been raised by the three reviewers, it may be worth expanding a bit more on the interpretation of these data. The fact that these origins are called b-catenin-induced is a bit misleading as it seems that they appear out of the blue when b-catenin is induced. As described in the ms, it is very likely that pre-RCs at these origins assemble in control cells and are wiped out by transcription during the long G1 phase. This process may be incomplete in some control cells, leading to a weak activation of these origins.

We, the authors, are grateful to the Referees for their constructive comments on our manuscript. We have addressed all the points raised by the Referees, which significantly improved our manuscript.

Point-by-point response:

Reviewer #2:

Overall, the manuscript addresses an important topic and uses a highly innovative methodology to investigate the relation of oncogene expression and defective HR. A comment that I originally raised, and which is only partially addressed, is that the conclusions and title are too generic, and should be toned down and be phrased more specifically to match the results of the experiments.

The far majority of the results reflect b-catenin upregulation, as also reflected in the final model that the authors present. I would highly recommend adjusting the title, abstract and start of the discussion accordingly, and mention that parts of these effects can be extended to other oncogenes.

Comments:

-The title and abstract state that oncogene amplification is mutually exclusive with BRCA1/2 inactivation. This is true only for the selected panel of oncogenes. The authors in their rebuttal letter indicate that MYC is not mutually exclusive with BRCA1/2 mutation. Of note, previous work by the authors showed that MYC and Cyclin E both trigger aberrant origin firing (Macharet, Nature, 2018). So, the conclusion cannot be that this is a generic mechanism explaining SL interactions between oncogene activation and BRCA defects. Although the authors acknowledge this is in the rebuttal letter, these data are not discussed in the manuscript or toned down the conclusions.

Also, the first part of the discussion still describes the results as a universal mechanism. To present a picture that does justice to the complexity of HR-defective tumors, the above-mentioned data should be part of the figures and discussion in the manuscript, not only the rebuttal letter.

I realize that text is added to the discussion on MYC. However, this part only provides a very limited interpretation of the available literature. In the updated discussion, Myc is described as an exceptional case, which works through triggering copy number variations, including MCL1. However, similar long-term effects driving CNA acquisition have been described for CCNE1 (eg. Negrini, 2010 and Aziz, 2019), which is not described to discard the model. More importantly, Myc has been shown to triggers de novo origin firing like CCNE1, in short-term experiments. Moreover, Myc is not only regulated at the transcriptional level by b-catenin, but is also amplified frequently, and is not mutual exclusive with BRCA1/2 mutations in tumor databases.

Response: We have altered the title and abstract to reflect that the majority of our results are due to β -catenin accumulation.

-concerning the lethal effects of oncogene expression in BRCA1/2 defective cells, Cyclin D and cyclin E are now included (Suppl. Figure 4), overexpression of which clearly induces S-phase entry, both in control and BRCA2-depleted cells.

The experiment in Suppl. 4c/d is difficult to interpret when the effects are only plotted as a separate relative measure in subcategories. BRCA1/2 inactivation is lethal in many cell lines, and these effects cannot be judged in these plots. These plots should ideally be visualized as absolute proliferation measures. Alternatively, the plots could be plotted as relative measures to only one condition (-dox/siControl) to appreciate all relevant effects in these experiments.

Also, the interpretation of the experiment should be toned down. Lower proliferation is not the same as viability or toxicity (Line 187/188: 'specifically toxic'; Line 190: 'decrease in viability', Line 479: 'the same toxicity';) but should be referred to as 'decreased proliferation'.

Response: We believe that our display makes it easier for the reader to interpret the effect of cyclin E or cyclin D1 overexpression on BRCA-depleted cells. We have altered the text to remove viability and toxicity and replaced with decreased proliferation.

-a minor, but important issue: the labeling of the figures contains an interpretation of the experiment, rather than the actual procedure. '-BRCA2' or '+BRCA2' could indicate reconstitution, knockout etc, but in this situation reflects a dox-inducible shRNA. '+/- dox' is much closer to the actual experiments (like in Fig1e, where siControl and siB-catenin is used properly, as well as in figures of Suppl. 4, where +/- dox is used). The same is true for 1C, where 'BRCA2-proficient' vs 'BRCA2-deficient' is used.

Response: We believe that our data labelling style (+/-BRCA2, rather than -/+DOX) makes it easier for the reader to follow our manuscript. We therefore kept the existing labels and we included in all figure legends the explanation that +/-BRCA2 labels indicate -/+DOX.

-line 134: 'oncogenic B-catenin activation' refers to treatment with a GSK3 inhibitor. It remains unclear if this reflects 'oncogenic' B-catenin activation. The correct way to state this would be: 'GSK3 inhibition is lethal with BRCA1 or BRCA2 inactivation' or 'b-catenin activation is lethal with BRCA1 or BRCA2 inactivation'. Similarly: 'its oncogenic expression' should be 'its expression'. Also applies to 'oncogenic' in titles of suppl. Fig 4 and 5.

Response: Previous literature has established that stabilisation of β -catenin leads to its oncogenic activation (reviewed in Polakis P., Curr Opin Genet Dev 1999) we therefore kept 'oncogenic' in the text.

-suppl. Figure 2c/e: BRCA2 blots are lacking, which are needed to interpret the experiment.

Response: The cells from this figure were used in Reislander et al. (Nat Comm 2019) where the BRCA2 blots were published.

-line 1346: SEM cannot be derived from a single experiment. I guess this should be 'SEM of the three independent experiments'.

Response: We have removed the error bars to reflect that this graph is the mean of technical triplicates from one experiment.

Reviewer #3:

The authors have significantly improved their manuscript and have addressed all the issues that I raised. In my opinion, this work is now suitable for publication in Nature Communications. Since the question of why intragenic oncogene-induced origins are active in control cells (albeit at a lower level) has been raised by the three reviewers, it may be worth expanding a bit more on the interpretation of these data. The fact that these origins are called b-catenin-induced is a bit misleading as it seems that they appear out of the blue when b-catenin is induced. As described in the ms, it is very likely that pre-RCs at these origins assemble in control cells and are wiped out by transcription during the long G1 phase. This process may be incomplete in some control cells, leading to a weak activation of these origins.

Response: We have extended the interpretation of these results in the Discussion section. We detailed that illegitimate intragenic origin firing is rarely/weakly detected in control cells but strongly induced by β -catenin accumulation. We mentioned that whilst transcription-mediated origin inactivation during G1 inactivate intragenic origins, failure to complete this process in a subset of control cells leads to the is rarely/weakly detection of illegitimate intragenic origin firing even in the absence of by β -catenin accumulation.